# Towards Unifying Hamiltonian Monte Carlo and Slice Sampling

**Yizhe Zhang, Xiangyu Wang, Changyou Chen, Ricardo Henao, Kai Fan, Lawrence Carin**
Duke University
Durham, NC, 27708
{yz196,xw56,changyou.chen, ricardo.henao, kf96 , lcarin} @duke.edu

## Abstract

We unify slice sampling and Hamiltonian Monte Carlo (HMC) sampling, demonstrating their connection via the Hamiltonian-Jacobi equation from Hamiltonian mechanics. This insight enables extension of HMC and slice sampling to a broader family of samplers, called Monomial Gamma Samplers (MGS). We provide a theoretical analysis of the mixing performance of such samplers, proving that in the limit of a single parameter, the MGS draws decorrelated samples from the desired target distribution. We further show that as this parameter tends toward this limit, performance gains are achieved at a cost of increasing numerical difficulty and some practical convergence issues. Our theoretical results are validated with synthetic data and real-world applications.

## 1 Introduction

Markov Chain Monte Carlo (MCMC) sampling [1] stands as a fundamental approach for probabilistic inference in many computational statistical problems. In MCMC one typically seeks to design methods to efficiently draw samples from an unnormalized density function. Two popular auxiliary-variable sampling schemes for this task are Hamiltonian Monte Carlo (HMC) [2, 3] and the slice sampler [4]. HMC exploits gradient information to propose samples along a trajectory that follows *Hamiltonian dynamics* [3], introducing momentum as an auxiliary variable. Extending the random proposal associated with Metropolis-Hastings sampling [4], HMC is often able to propose large moves with acceptance rates close to one [2]. Recent attempts toward improving HMC have leveraged geometric manifold information [5] and have used better numerical integrators [6]. Limitations of HMC include being sensitive to parameter tuning and being restricted to continuous distributions. These issues can be partially solved by using adaptive approaches [7, 8], and by transforming sampling from discrete distributions into sampling from continuous ones [9, 10].

Seemingly distinct from HMC, the slice sampler [4] alternates between drawing conditional samples based on a target distribution and a uniformly distributed slice variable (the auxiliary variable). One problem with the slice sampler is the difficulty of solving for the slice interval, *i.e.*, the domain of the uniform distribution, especially in high dimensions. As a consequence, adaptive methods are often applied [4]. Alternatively, one recent attempt to perform efficient slice sampling on latent Gaussian models samples from a high-dimensional elliptical curve parameterized by a single scalar [11]. It has been shown that in some cases slice sampling is more efficient than Gibbs sampling and Metropolis-Hastings, due to the adaptability of the sampler to the scale of the region currently being sampled [4].

Despite the success of slice sampling and HMC, little research has been performed to investigate their connections. In this paper we use the Hamilton-Jacobi equation from classical mechanics to show that slice sampling is equivalent to HMC with a (simply) *generalized* kinetic function. Further, we also show that different settings of the HMC kinetic function correspond to *generalized* slice

sampling, with a *non-uniform* conditional slicing distribution. Based on this relationship, we develop theory to analyze the newly proposed broad family of auxiliary-variable-based samplers. We prove that under this special family of distributions for the momentum in HMC, as the distribution becomes more heavy-tailed, the one-step autocorrelation of samples from the target distribution converges *asymptotically* to zero, leading to potentially decorrelated samples. While of limited *practical* impact, this theoretical result provides insights into the properties of the proposed family of samplers. We also elaborate on the practical tradeoff between the increased computational complexity associated with improved theoretical sampling efficiency. In the experiments, we validate our theory on both synthetic data and with real-world problems, including Bayesian Logistic Regression (BLR) and Independent Component Analysis (ICA), for which we compare the mixing performance of our approach with that of standard HMC and slice sampling.

## 2 Solving Hamiltonian dynamics via the Hamilton-Jacobi equation

A Hamiltonian system consists of a *kinetic* function $K(p)$ with *momentum* variable $p \in \mathbb{R}$, and a potential energy function $U(x)$ with coordinate $x \in \mathbb{R}$. We elaborate on multivariate cases in the Appendix. The dynamics of a Hamiltonian system are completely determined by a set of first-order Partial Differential Equations (PDEs) known as *Hamilton's equations* [12]:

$$\frac{\partial p}{\partial \tau} = -\frac{\partial H(x, p, \tau)}{\partial x} , \qquad \frac{\partial x}{\partial \tau} = \frac{\partial H(x, p, \tau)}{\partial p} , \qquad (1)$$

where $H(x, p, \tau) = K(p(\tau)) + U(x(\tau))$ is the *Hamiltonian*, and $\tau$ is the system time. Solving (1) gives the dynamics of $x(\tau)$ and $p(\tau)$ as a function of system time $\tau$. In a Hamiltonian system governed by (1), $H(\cdot)$ is a constant for every $\tau$ [12]. A specified $H(\cdot)$, together with the initial point $\{x(0), p(0)\}$, defines a *Hamiltonian trajectory* $\{\{x(\tau), p(\tau)\} : \forall \tau\}$, in $\{x, p\}$ space.

It is well known that in many practical cases, a direct solution to (1) may be difficult [13]. Alternatively, one might seek to transform the original HMC system $\{H(\cdot), x, p, \tau\}$ to a dual space $\{H'(\cdot), x', p', \tau\}$ in hope that the transformed PDEs in the dual space becomes simpler than the original PDEs in (1). One promising approach consists of using the *Legendre* transformation [12]. This family of transformations defines a unique mapping between primed and original variables, where the system time, $\tau$, is identical. In the transformed space, the resulting dynamics are often simpler than the original Hamiltonian system.

An important property of the *Legendre* transformation is that the form of (1) is preserved in the new space [14], *i.e.*, $\partial p'/\partial \tau = -\partial H'(x', p', \tau)/\partial x'$, $\partial x'/\partial \tau = \partial H'(x', p', \tau)/\partial p'$. To guarantee a valid *Legendre* transformation between the original Hamiltonian system $\{H(\cdot), x, p, \tau\}$ and the transformed Hamiltonian system $\{H'(\cdot), x', p', \tau\}$, both systems should satisfy the *Hamilton's principle* [13], which equivalently express Hamilton's equations (1). The form of this *Legendre* transformation is not unique. One possibility is to use a generating function approach [13], which requires the transformed variables to satisfy $p \cdot \partial x/\partial \tau - H(x, p, \tau) = p' \cdot \partial x'/\partial \tau - H(x', p', \tau)' + dG(x, x', p', \tau)/d\tau$, where $dG(x, x', p', \tau)/d\tau$ follows from the chain rule and $G(\cdot)$ is a Type-2 generating function defined as $G(\cdot) \triangleq -x' \cdot p' + S(x, p', \tau)$ [14], with $S(x, p', \tau)$ being the *Hamilton's principal function* [15], defined below. The following holds due to the independency of $x$, $x'$ and $p'$ in the previous transformation (after replacing $G(\cdot)$ by its definition):

$$p = \frac{\partial S(x, p', \tau)}{\partial x} , \qquad x' = \frac{\partial S(x, p', \tau)}{\partial p'} , \qquad H'(x', p', \tau) = H(x, p, \tau) + \frac{\partial S(x, p', \tau)}{\partial \tau} . \quad (2)$$

We then obtain the desired *Legendre* transformation by setting $H'(x', p', \tau) = 0$. The resulting (2) is known as the *Hamilton-Jacobi equation* (HJE). We refer the reader to [13, 12] for extensive discussions on the *Legendre* transformation and HJE.

Recall from above that the *Legendre* transformation preserves the form of (1). Since $H'(x', p', \tau) = 0$, $\{x', p'\}$ are *time-invariant* (constant for every $\tau$). Importantly, the *time-invariant* point $\{x', p'\}$ corresponds to a Hamiltonian *trajectory* in the original space, and it defines the initial point $\{x(0), p(0)\}$ in the original space $\{x, p\}$; hence, given $\{x', p'\}$, one may update the point along the trajectory by specifying the time $\tau$. A new point $\{x(\tau), p(\tau)\}$ in the original space along the Hamiltonian trajectory, with system time $\tau$, can be determined from the transformed point $\{x', p'\}$ via solving (2).

One typically specifies the kinetic function as $K(p) = p^2$ [2], and Hamilton's principal function as $S(x, p', \tau) = W(x) - p'\tau$, where $W(x)$ is a function to be determined (defined below). From (2),

and the definition of $S(\cdot)$, we can write

$$H(x, p, \tau) + \frac{\partial S}{\partial \tau} = H(x, p, \tau) - p' = U(x) + \left[\frac{\partial S}{\partial x}\right]^2 - p' = U(x) + \left[\frac{dW(x)}{dx}\right]^2 - p' = 0 , \quad (3)$$

where the second equality is obtained by replacing $H(x, p, \tau) = U(x(\tau)) + K(p(\tau))$ and the third equality by replacing $p$ from (2) into $K(p(\tau))$. From (3), $p' = H(x, p, \tau)$ represents the total Hamiltonian in the original space $\{x, p\}$, and uniquely defines a Hamiltonian trajectory in $\{x, p\}$.

Define $\mathbb{X} \triangleq \{x : H(\cdot) - U(x) \geq 0\}$ as the *slice interval*, which for constant $p' = H(x, p, \tau)$ corresponds to a set of valid coordinates in the original space $\{x, p\}$. Solving (3) for $W(x)$ gives

$$W(x) = \int_{x_{min}}^{x(\tau)} f(z)^{\frac{1}{2}} dz + C , \qquad f(z) = \begin{cases} H(\cdot) - U(z), & z \in \mathbb{X} \\ 0, & z \notin \mathbb{X} \end{cases} , \quad (4)$$

where $x_{min} = \min\{x : x \in \mathbb{X}\}$ and $C$ is a constant. In addition, from (2) we have

$$x' = \frac{\partial S(x, p', \tau)}{\partial p'} = \frac{\partial W(x)}{\partial H} - \tau = \frac{1}{2} \int_{x_{min}}^{x(\tau)} f(z)^{-\frac{1}{2}} dz - \tau , \quad (5)$$

where the second equality is obtained by substituting $S(\cdot)$ by its definition and the third equality is obtained by applying Fubini's theorem on (4). Hence, for constant $\{x', p' = H(x, p, \tau)\}$, equation (5) *uniquely* defines $x(\tau)$ in the original space, for a specified system time $\tau$.

## 3 Formulating HMC as a Slice Sampler

### 3.1 Revisiting HMC and Slice Sampling

Suppose we are interested in sampling a random variable $x$ from an unnormalized density function $f(x) \propto \exp[-U(x)]$, where $U(x)$ is the potential energy function. *Hamiltonian Monte Carlo* (HMC) augments the target density with an auxiliary momentum random variable $p$, that is independent of $x$. The distribution of $p$ is specified as $\propto \exp[-K(p)]$, where $K(p)$ is the kinetic energy function. Define $H(x, p) = U(x) + K(p)$ as the Hamiltonian. We have omitted the dependency of $H(\cdot)$, $x$ and $p$ on the system time $\tau$ for simplicity. HMC iteratively performs *dynamic evolving* and *momentum resampling* steps, by sampling $x_t$ from the target distribution and $p_t$ from the momentum distribution (Gaussian as $K(p) = p^2$), respectively, for $t = 1, 2, \ldots$ iterations. Figure 1 illustrates two iterations of this procedure. Starting

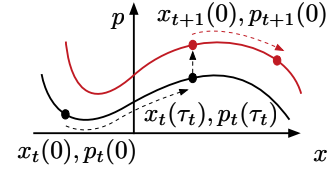

Figure 1: Representation of HMC sampling. Points $\{x_t(0), p_t(0)\}$ and $\{x_{t+1}(0), p_{t+1}(0)\}$ represent HMC samples at iterations $t$ and $t + 1$, respectively. The trajectories for $t$ and $t + 1$ correspond to distinct Hamiltonian levels $H_t(\cdot)$ and $H_{t+1}(\cdot)$, denoted as black and red lines, respectively.

from point $\{x_t(0), p_t(0)\}$ at the $t$-th (discrete) iteration, HMC leverages the Hamiltonian dynamics, governed by *Hamilton's equations* in (1) to propose the next sample $\{x_t(\tau_t), p_t(\tau_t)\}$, at system time $\tau_t$. The *position* in HMC at iteration $t + 1$ is updated as $x_{t+1}(0) = x_t(\tau_t)$ (*dynamic evolving*). A new momentum $p_{t+1}(0)$ is resampled independently from a Gaussian distribution (assuming $K(p) = p^2$), establishing the next initial point $\{x_{t+1}(0), p_{t+1}(0)\}$ for iteration $t + 1$ (*momentum resampling*). The latter point corresponds to the initial point of a new trajectory because the Hamiltonian $H(\cdot)$ is commensurately updated. This means that trajectories correspond to distinct values of $H(\cdot)$.

Typically, numerical integrators such as the *leap-frog* method [2] are employed to numerically approximate the Hamiltonian dynamics. In practice, a random number (uniformly drawn from a fixed range) of discrete numerical integration steps (leap-frog steps) are often used (corresponding to random time $\tau_t$ along the trajectory), which has been shown to have better convergence properties than a single leap-frog step [16]. The discretization error introduced by the numerical integration is corrected by a Metropolis Hastings (MH) step.

*Slice sampling* is conceptually simpler than HMC. It augments the target unnormalized density $f(x)$ with a random variable $y$, with joint distribution expressed as $p(x, y) = Z_1^{-1}$, s.t. $0 < y < f(x)$, where $Z_1 = \int f(x) dx$ is the normalization constant, and the marginal distribution of $x$ exactly recovers the target normalized distribution $f(x)/Z_1$. To sample from the target density, slice sampling iteratively performs a *conditional sampling step* from $p(x|y)$ and *sampling a slice* from $p(y|x)$. At iteration $t$, starting from $x_t$, a slice $y_t$ is uniformly drawn from $(0, f(x_t))$. Then, the next sample $x_{t+1}$, at iteration $t + 1$, is uniformly drawn from the *slice interval* $\{x : f(x) > y_t\}$.

HMC and slice sampling both augment the target distribution with *auxiliary variables* and can propose long-range moves with high acceptance probability.

## 3.2 Formulating HMC as a Slice Sampler

Consider the *dynamic evolving* step in HMC, *i.e.*, $\{x_t(0), p_t(0)\} \mapsto \{x_t(\tau), p_t(\tau)\}$ in Figure 1. From Section 2, the Hamiltonian dynamics in $\{x, p\}$ space with initial point $\{x(0), p(0)\}$ can be performed by mapping to $\{x', p'\}$ space and updating $\{x(\tau), p(\tau)\}$ via selecting a $\tau$ and solving (5). As we show in the Appendix, from (5) and in univariate cases* the Hamiltonian dynamics has period $\int_{\mathbb{X}} [H(\cdot) - U(z)]^{-\frac{1}{2}} dz$ and is symmetric along $p = 0$ (due to the symmetric form of the kinetic function). Also from (5), the system time, $\tau$, is specified uniformly sampled from a half-period of the Hamiltonian dynamics. *i.e.*, $\tau \sim \text{Uniform}\left(-x', -x' + \frac{1}{2} \int_{\mathbb{X}} [H(\cdot) - U(z)]^{-\frac{1}{2}}\right)$. Intuitively, $x'$ is the "anchor" of the initial point $\{x(0), p(0)\}$, w.r.t. the start of the first half period, *i.e*, when $\int_{\mathbb{X}} [H(\cdot) - U(z)]^{-\frac{1}{2}} = 0$. Further, we only need consider half a period because for a symmetric kinetic function, $K(p) = p^2$, the Hamiltonian dynamics for the two half-periods are mirrored [14]. For the same reason, Figure 1 only shows half of the $\{x, p\}$ space, when $p \geq 0$.

Given the sampled $\tau$ and the constant $\{x', p'\}$, equation (5) can be solved for $x^* \triangleq x(\tau)$, *i.e.*, the value of $x$ at time $\tau$. Interestingly, the integral in (5) can be interpreted as (up to normalization constant) a cumulative density function (CDF) of $x(\tau)$. From the inverse CDF transform sampling method, uniformly sampling $\tau$ from half of a period and solving for $x^*$ from (5), are equivalent to directly sampling $x^*$ from the following density

$$p(x^*|H(\cdot)) \propto [H(\cdot) - U(x^*)]^{-\frac{1}{2}}, \qquad \text{s.t., } H(\cdot) - U(x^*) \geq 0. \qquad (6)$$

We note that this transformation does not make the analytic solution of $x(\tau)$ generally tractable. However, it provides the basic setup to reveal the connection between the slice sampler and HMC.

In the *momentum resampling* step of HMC, *i.e.*, $\{x_t(\tau), p_t(\tau)\} \mapsto \{x_{t+1}(0), p_{t+1}(0)\}$ in Figure 1, and using the previously described kinetic function, $K(p) = p^2$, resampling corresponds to drawing $p$ from a Gaussian distribution [2].

The algorithm to analytically sample from the HMC (*analytic HMC*) proceeds as follows: at iteration $t$, momentum $p_t$ is drawn from a Gaussian distribution. The previously sampled value of $x_{t-1}$ and the newly sampled $p_t$ yield a Hamiltonian $H_t(\cdot)$. Then, the next sample $x_t$ is drawn from (6). This procedure relates HMC to the slice sampler. To clearly see the connection, we denote $y_t = e^{-H_t(\cdot)}$. Instead of directly sampling $\{p, x\}$ as just described, we sample $\{y, x\}$ instead. By substituting $H_t(\cdot)$ with $y_t$ in (6), the conditional updates for this new sampling procedure can be rewritten as below, yielding the *HMC slice sampler* (HMC-SS), with conditional distributions defined as

$$\text{Sampling a slice:} \quad p(y_t|x_t) = \frac{1}{\Gamma(a) f(x_t)} [\log f(x_t) - \log y_t]^{1-a}, \quad \text{s.t. } 0 < y_t < f(x_t), \quad (7)$$

$$\text{Conditional sampling:} \quad p(x_{t+1}|y_t) = \frac{1}{Z_2(y_t)} [\log f(x_{t+1}) - \log y_t]^{1-a}, \quad \text{s.t. } f(x_t) > y_t, \quad (8)$$

where $a = 1/2$ (other values of $a$ considered below), $f(x) = e^{-U(x)}$ is an unnormalized density, and $Z_1 \triangleq \int f(x) dx$ and $Z_2(y) \triangleq \int_{f(x)>y} [\log f(x) - \log y]^{-\frac{1}{2}} dx$ are the normalization constants.

Comparing these two procedures, analytic HMC and HMC-SS, we see that the *resampling momentum* in analytic HMC corresponds to *sampling a slice* in HMC-SS. Further, the *dynamic evolving* in HMC corresponds to the *conditional sampling* in MG-SS. We have thus shown that HMC can be equivalently formulated as a slice sampler procedure via (7) and (8).

## 3.3 Reformulating Standard Slice Sampler from HMC-SS

In *standard* slice sampling (described in Section 3.1), both conditional sampling and sampling a slice are drawn from *uniform* distributions. However those for HMC-SS in (7) and (8) represent *non-uniform* distributions. Interestingly, if we change $a$ in (7) and (8) from $a = 1/2$ to $a = 1$, we obtain the desired uniform distributions for standard slice sampling. This key observation leads us to consider a generalized form of the kinetic function for HMC, described below.

Consider the *generalized* family of kinetic functions $K(p) = |p|^{1/a}$ with $a > 0$. One may rederive equations (3)-(8) using this generalized kinetic energy. As shown in the Appendix, these equations remained unchanged, with the update that each isolated 2 in these equations is replaced by $1/a$, and $-1/2$ is replaced by $a - 1$.

Sampling $p$ (for the *momentum resampling* step) with the generalized kinetics, corresponds to drawing $p$ from $\pi(p; m, a) = \frac{1}{2} m^{-a} / \Gamma(a + 1) \exp[-|p|^{1/a}/m]$, with $m = 1$. All the formulation in the paper still holds for arbitrary $m$, see Appendix for details. We denote this distribution the *monomial Gamma* (MG) distribution, $\mathrm{MG}(a, m)$, where $m$ is the *mass parameter*, and $a$ is the *monomial parameter*. Note that this is equivalent to the exponential power distribution with zero-mean, described in [17]. We summarize some properties of the MG distribution in the Appendix.

To generate random samples from the MG distribution, one can draw $G \sim \mathrm{Gamma}(a, m)$ and a uniform sign variable $S \sim \{-1, 1\}$, then $S \cdot G^a$ follows the $\mathrm{MG}(a, m)$ distribution. We call the HMC sampler based on the generalized kinetic function, $K(p; a, m)$: *Monomial Gamma Hamiltonian Monte Carlo* (MG-HMC). The algorithm to analytically sample from the MG-HMC is shown in Algorithm 1. The only difference between this procedure and the previously described is the momentum resampling step, in that for *analytic HMC*, $p$ is drawn Gaussian instead of $\mathrm{MG}(a, m)$. However, note that the Gaussian distribution is a special case of $\mathrm{MG}(a, m)$ when $a = 1/2$.

| **Algorithm 1:** MG-HMC with HJE | **Algorithm 2:** MG-SS |
|---|---|
| **for** $t = 1$ *to* $T$ **do**<br>  Resample momentum: $p_t \sim \mathrm{MG}(m, a)$.<br>  Compute Hamiltonian: $H_t = U(x_{t-1}) + K(p_t)$.<br>  Find $\mathbb{X} \triangleq \{x : x \in \mathbb{R}; U(x) \le H_t(\cdot)\}$.<br>  Dynamic evolving: $x_t | H_t(\cdot) \propto [H_t(\cdot) - U(x_t)]^{a-1}$ ; $x \in \mathbb{X}$. | **for** $t = 1$ *to* $T$ **do**<br>  Sampling a slice:<br>  Sample $y_t$ from (7).<br>  Conditional sampling:<br>  Sample $x_t$ from (8). |

Interestingly, when $a = 1$, the *Monomial Gamma Slice sampler* (MG-SS) in Algorithm 2 recovers exactly the same update formulas as in standard slice sampling, described in Section 3.1, where the conditional distributions in (7) and (8) are both uniform. When $a \ne 1$, we have to iteratively alternate between sampling from non-uniform distributions (7) and (8), for both auxiliary (slicing) variable $y$ and target variable $x$.

Using the same argument from the convergence analysis of standard slice sampling [4], the iterative sampling procedure in (7) and (8), converges to an invariant joint distribution (detailed in the Appendix). Further, the marginal distribution of $x$ recovers the target distribution as $f(x)/Z_1$, while the marginal distribution of $y$ is given by $p(y) = Z_2(y)/[\Gamma(a)Z_1]$.

The MG-SS can be divided into three broad regimes: $0 < a < 1, a = 1$ and $a > 1$ (illustrated in the Appendix). When $0 < a < 1$, the conditional distribution $p(y_t|x_t)$ is skewed towards the current unnormalized density value $f(x_t)$. The conditional draw of $p(x_{t+1}|y_t)$ encourages taking samples with smaller density value (inefficient moves), within the domain of the slice interval $\mathbb{X}$. On the other hand, when $a > 1$, draws of $y_t$ tend to take smaller values, while draws of $x_{t+1}$ encourage sampling from those with large density function values (efficient moves). The case $a = 1$ corresponds to the conventional slice sampler. Intuitively, setting $a$ to be small makes the auxiliary variable, $y_t$, stay close to $f(x_t)$, thus $f(x_{t+1})$ is close to $f(x_t)$. As a result, a larger $a$ seems more desirable. This intuition is justified in the following sections.

## 4 Theoretical analysis

We analyze theoretical properties of the MG sampler. All the proofs as well as the ergodicity properties of analytic MG-SS are given in the Appendix.

**One-step autocorrelation of analytic MG-SS**  We present results on the univariate distribution case: $p(x) \propto e^{-U(x)}$. We first investigate the impact of the monomial parameter $a$ on the one-step *autocorrelation function* (ACF), $\rho_x(1) \triangleq \rho(x_t, x_{t+1}) = [\mathbb{E}x_t x_{t+1} - (\mathbb{E}x)^2]/\mathrm{Var}(x)$, as $a \to \infty$. Theorem 1 characterizes the limiting behavior of $\rho(x_t, x_{t+1})$.

**Theorem 1** *For a univariate target distribution,* i.e. $\exp[-U(x)]$ *has finite integral over* $\mathbb{R}$, *under certain regularity conditions, the one-step autocorrelation of the MG-SS parameterized by* $a$, *asymptotically approaches zero as* $a \to \infty$, i.e., $\lim_{a \to 0} \rho_x(1) = 0$.

In the Appendix we also show that $\lim_{a\to\infty}\rho(y_t,y_{t+1})=0$. In addition, we show that $\rho(y_t,y_{t+h})$ is a non-negative decreasing function of the time lag in discrete steps $h$.

**Effective sample size**    The variance of a Monte Carlo estimator is determined by its Effective Sample Size (ESS) [18], defined as $\text{ESS}=N/(1+2\times\sum_{h=1}^{\infty}\rho_x(h))$, where $N$ is the total number of samples, $\rho_x(h)$ is the $h$-step autocorrelation function, which can be calculated in a recursive manner. We prove in the Appendix that $\rho_x(h)$ is non-negative. Further, assuming the MG sampler is uniformly ergodic and $\rho_x(h)$ is monotonically decreasing, it can be shown that $\lim_{a\to\infty}\text{ESS}=N$. When ESS approaches full sample size, $N$, the resulting sampler delivers excellent mixing efficiency [5]. Details and further discussion are provided in the Appendix.

**Case study**    To examine a specific 1D example, we consider sampling from the exponential distribution, $\text{Exp}(\theta)$, with energy function given by $U(x)=x/\theta$, where $x\geq 0$. This case has analytic $\rho_x(h)$ and ESS. After some algebra (details in the Appendix),

$$\rho_x(1)=\frac{1}{a+1}\,,\ \rho_x(h)=\frac{1}{(a+1)^h}\,,\ \text{ESS}=\frac{Na}{a+2}\,,\hat{x}_h(x_0)\triangleq\mathbb{E}_{\kappa_h(x_h|x_0)}x_h=\theta+\frac{x_0-\theta}{(a+1)^h}.$$

These results are in agreement with Theorem 1 and related arguments of ESS and monotonicity of autocorrelation w.r.t. $a$. Here $\hat{x}_h(x_0)$ denotes the expectation of the $h$-lag sample, starting from any $x_0$. The relative difference $\frac{\hat{x}_h(x_0)-\theta}{x_0-\theta}$ decays exponentially in $h$, with a factor of $\frac{1}{a+1}$. In fact, the $\rho_x(1)$ for the exponential family class of models introduced in [19], with potential energy $U(x)=x^\omega/\theta$, where $x\geq 0,\omega,\theta>0$, can be analytically calculated. The result, provided in the Appendix, indicates that for this family, $\rho_x(1)$ decays at a rate of $\mathcal{O}(a^{-1})$.

**MG-HMC mixing performance**    In theory, the *analytic MG-HMC* (the dynamics in (5) can be solved exactly) is expected to have the same theoretical properties of the analytic MG-SS for *unimodal cases*, since they are derived from the same setup. However, the mixing performance of the two methods could differ significantly when sampling from a *multimodal* distribution, due to the fact that the Hamiltonian dynamics may get "trapped" into a single closed trajectory (one of the modes) with low energy, whereas the analytic MG-SS does not suffer from this problem as is able to sample from disjoint slice intervals (one per mode). This is a well-known property of slice sampling [4] that arises from (7) and (8). However, if $a$ is large enough, as we show in the Appendix, the probability of getting into a low-energy level associated with more than one Hamiltonian trajectory, which restrict movement between modes, is arbitrarily small. As a result, the analytic MG-HMC with large value of $a$ is able to approach the stationary mixing performance of MG-SS.

## 5   MG sampling in practice

**MG-HMC with numerical integrator**    In practice, MG-SS (performing Algorithm 2) requires: 1) analytically solving for the slice interval $\mathbb{X}$, which is typically infeasible for multivariate cases [4]; or 2) analytically computing the integral $Z_2(y)$ over $\mathbb{X}$, implied by the non-uniform conditionals from MG-SS. These are usually computationally infeasible, though adaptive estimation of $\mathbb{X}$ could be done using schemes like "doubling" and "shrinking" strategies from the slice sampling literature [4].

It is more convenient to perform approximate MG-HMC using a numerical integrator like in traditional HMC, *i.e.*, in each iteration, the momentum $p$ is first initialized by sampling from $\text{MG}(m,a)$, then second order Störmer-Verlet integration [2] is performed for the Hamiltonian dynamics updates:

$$\mathbf{p}_{t+1/2}=\mathbf{p}_t-\tfrac{\epsilon}{2}\nabla U(\mathbf{x}_t)\,,\ \ \mathbf{x}_{t+1}=\mathbf{x}_t+\epsilon\nabla K(\mathbf{p}_{t+1/2})\,,\ \ \mathbf{p}_{t+1}=\mathbf{p}_{t+1/2}-\tfrac{\epsilon}{2}\nabla U(\mathbf{x}_{t+1})\,,\quad(9)$$

where $\nabla K(\mathbf{p})=\text{sign}(\mathbf{p})\cdot\frac{1}{ma}|\mathbf{p}|^{1/a-1}$. When $a=1$, $[\nabla K(\mathbf{p})]_d=1/m$ for any dimension $d$, independent of $\mathbf{x}$ and $\mathbf{p}$. To avoid moving on a grid when $a=1$, we employ a random step-size $\epsilon$ from a uniform distribution within non-negative range $(r_1,r_2)$, as suggested in [2].

**No free lunch**    With a numerical integrator for MG-HMC, however, the argument about choosing large $a$ (of great theoretical advantage as discussed in the previous section) may face practical issues.

First, a large value of $a$ will lead to a less accurate numerical integrator. This is because as $a$ gets larger, the trajectory of the total Hamiltonian becomes "stiffer", *i.e.*, that the maximum curvature becomes larger. When $a>1/2$, the Hamiltonian trajectory in the phase space, $(\mathbf{x},\mathbf{p})$, has at least $2^D$ ($D$ denotes the total dimension) non-differentiable points ("turnovers"), at each intersection point with the hyperplane $\mathbf{p}^{(d)}=0,d\in\{1\cdots D\}$. As a result, directly applying Störmer-Verlet integration would lead to high integration error as $D$ becomes large.

Second, if the sampler is initialized in the tail region of a light-tailed target distribution, MG-HMC with $a > 1$ may converge arbitrarily slow to the true target distribution, *i.e.*, the burn-in period could take arbitrarily long time. For example, with $a > 1$, $\nabla U(x_0)$ can be very large when $x_0$ is in the light-tailed region, leading the update $x_0 + \nabla K(p_0 + \nabla U(x_0))$ to be arbitrary close to $x_0$, *i.e.*, the sampler does not move.

To ameliorate these issues, we provide mitigating strategies. For the first (numerical) issue, we propose two possibilities: 1) As an analog to the *"reflection"* action of [2], in (9), whenever the $d$-th dimension(s) of the momentum changes sign, we "recoil" the point of these dimension(s) to the previous iteration, and negate the momentum of these dimension(s), *i.e.*, $\mathbf{x}_{t+1}^{(d)} = \mathbf{x}_t^{(d)}, \mathbf{p}_{t+1}^{(d)} = -\mathbf{p}_t^{(d)}$. 2) Substituting the kinetic function $K(\mathbf{p})$ with a *"softened"* kinetic function, and use importance sampling to sample the momentum. The details and comparison between the *"reflection"* action and *"softened"* kinetics are discussed in the Appendix.

For the second (convergence) issue, we suggest using a step-size decay scheme, *e.g.*, $\epsilon = \max(\epsilon_1 \rho^t, \epsilon_0)$. In our experiments we use $(\epsilon_1, \rho) = (10^6, 0.9)$, where $\epsilon_0$ is problem-specific. This approach empirically alleviates the slow convergence problem, however we note that a more principled way would be adaptively selecting $a$ during sampling, which is left for further investigation.

As a compromise between theoretical gains and practical issues, we suggest setting $a = 1$ (HMC implementation of a slice sampler) when the dimension is relatively large. This is because in our experiments, when $a > 1$, numerical errors and convergence issues tend to overwhelm the theoretical mixing performance gains described in Section 4.

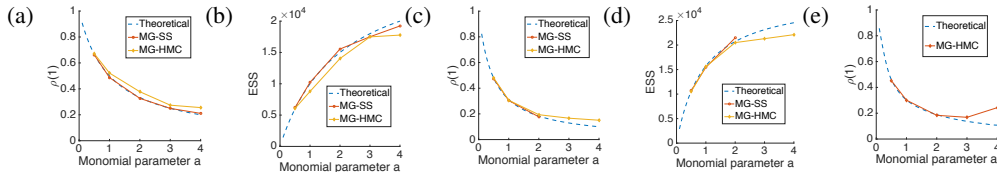

Figure 2: Theoretical and empirical $\rho_x(1)$ and ESS of exponential distribution (a,b), $\mathcal{N}_+$ (c,d) and Gamma (e).

## 6  Experiments

### 6.1  Simulation studies

**1D unimodal problems**  We first evaluate the performance of the MG sampler with several univariate distributions: 1) Exponential distribution, $U(x) = \theta x, x \geq 0$. 2) Truncated Gaussian, $U(x) = \theta x^2, x \geq 0$. 3) Gamma distribution, $U(x) = -(r-1)\log x + \theta x$. Note that the performance of the sampler does not depend on the scale parameter $\theta > 0$. We compare the empirical $\rho_x(1)$ and ESS of the analytic MG-SS and MG-HMC with their theoretical values. In the Gamma distribution case, analytic derivations of the autocorrelations and ESS are difficult, thus we resort to a numerical approach to compute $\rho_x(1)$ and ESS. Details are provided in the Appendix. Each method is run for 30,000 iterations with 10,000 burn-in samples. The number of leap-frog steps is set to be uniformly drawn from $(100 - l, 100 + l)$ with $l = 20$, as suggested by [16]. We also compared MG-HMC ($a = 1$) with standard slice sampling using doubling and shrinking scheme [4] As expected, the resulting ESS (not shown) for these two methods is almost identical. The experiment settings and results are provided in the Appendix. The acceptance rates decrease from around $0.98$ to around $0.77$ for each case, when $a$ grows from $0.5$ to $4$, as shown in Figure 2(a)-(d),

The results for analytic MG-SS match well with the theoretical results, however MG-HMC seems to suffer from practical difficulties when $a$ is large, evidenced by results gradually deviating from the theoretical values. This issue is more evident in the Gamma case (see Figure 2(e)), where $\rho_x(1)$ first decreases then increases. Meanwhile, the acceptance rates decreases from $0.9$ to $0.5$.

**1D and 2D bimodal problems**  We further conduct simulation studies to evaluate the efficiency of MG-HMC when sampling 1D and 2D multimodal distributions. For the univariate case, the potential energy is given by $U(x) = x^4 - 2x^2$; whereas $U(\mathbf{x}) = -0.2 \times (x_1 + x_2)^2 + 0.01 \times (x_1 + x_2)^4 - 0.4 \times (x_1 - x_2)^2$ in the bivariate case. We show in the Appendix that if the energy functions are symmetric along $\mathbf{x} = C$, where $C$ is a constant, in theory, the analytic MG-SS will have ESS equal to the total sample size. However, as shown in Section 4, the analytic MG-HMC is expected to have an ESS less than its corresponding analytic MG-SS, and the gap between the analytic MG-HMC

and analytic MG-SS counterpart should decrease with $a$. As a result, despite numerical difficulties, we expect the MG-HMC based on numerical integration to have better mixing performance with large $a$. To verify our theory, we run MG-HMC for $a = \{0.5, 1, 2\}$ for 30,000 iterations with 10,000 burn-in samples. The parameter settings and the acceptance rates are detailed in the Appendix. Empirically, we find that the efficiency of HMC is significantly improved with a large $a$ as shown in Table 1, which coincides with the theory in Section 4. From Figure 3, we observe that the MG-HMC sampler with monomial parameter $a = \{1, 2\}$ performs better at jumping between modes of the target distribution, when compared to standard HMC, which confirms the theory in Section 4. We also compared MG-HMC ($a = 1$) with standard SS [4]. As expected, in the 1D case, the standard SS yields ESS close to full sample size, while in 2D case, the resulting ESS is lower than MG-HMC ($a = 1$) (details are provided in the Appendix).

## 6.2 Real data

**Bayesian logistic regression** We evaluate our methods on 6 real-world datasets from the UCI repository [20]: German credit (G), Australian credit (A), Pima Indian (P), Heart (H), Ripley (R) and Caravan (C) [21]. Feature dimensions range from 7 to 87, and total data instances are between 250 to 5822. All datasets are normalized to have zero mean and unit variance. Gaussian priors $\mathcal{N}(\mathbf{0}, 100\mathbf{I})$

Figure 3: 10 MC samples by MG-HMC from a 2D distribution and different $a$.

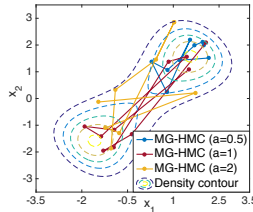

Table 1: ESS of MG-HMC for 1D and 2D bimodal distributions.

| 1D | ESS | $\rho_x(1)$ |
|---|---|---|
| $a = 0.5$ | 5175 | 0.60 |
| $a = 1$ | 10157 | 0.43 |
| $a = 2$ | 24298 | 0.11 |
| 2D | ESS | $\rho_x(1)$ |
| $a = 0.5$ | 4691 | 0.67 |
| $a = 1$ | 16349 | 0.60 |
| $a = 2$ | 18007 | 0.53 |

are imposed on the regression coefficients. We draw 5000 iterations with 1000 burn-in samples for each experiment. The leap-frog steps are set to be uniformly drawn from $(100 - l, 100 + l)$ with $l = 20$. Other experimental settings ($m$ and $\epsilon$) are provided in the Appendix.

Results in terms of minimum ESS are summarized in Table 2. Prediction accuracies estimated via cross-validation are almost identical all across (reported in the Appendix). It can be seen that MG-HMC with $a = 1$ outperforms (in terms of ESS) the other two settings with $a = 0.5$ and $a = 2$, indicating increased numerical difficulties counter the theoretical gains when $a$ becomes large. This can be also seen by noting that the acceptance rates drop from around 0.9 to around 0.7 as $a$ increases from 0.5 to 2. The dimensionality also seems to have an impact on the optimal setting of $a$, since in the high-dimensional dataset Cavaran, the improvement of MG-HMC with $a = 1$ is less significant compared with other datasets, and $a = 2$ seems to suffer more of numerical difficulties. Comparisons between MG-HMC ($a = 1$) and standard slice sampling are provided in the Appendix. In general, standard slice sampling with adaptive search underperforms relative to MG-HMC ($a = 1$).

Table 2: Minimum ESS for each method (dimensionality indicated in parenthesis). Left: BLR; Right: ICA

| Dataset (dim) | A (15) | G (25) | H (14) | P (8) | R (7) | C (87) | ICA (25) |
|---|---|---|---|---|---|---|---|
| $a = 0.5$ | 3124 | 3447 | 3524 | 3434 | 3317 | 33 (median 3987) | 2677 |
| $a = 1$ | **4308** | **4353** | **4591** | **4664** | **4226** | **36** (median 4531) | **3029** |
| $a = 2$ | 1490 | 3646 | 4315 | 4424 | 1490 | 7 (median 740) | 1534 |

**ICA** We finally evaluate our methods on the MEG [22] dataset for Independent Component Analysis (ICA), with 17,730 time points and 25 feature dimension. All experiments are based on 5000 MCMC samples. The acceptance rates for $a = (0.5, 1, 2)$ are $(0.98, 0.97, 0.77)$. Running time is almost identical for different $a$. Settings (including $m$ and $\epsilon$) are provided in the Appendix. As shown in Table 2, when $a = 1$, MG-HMC has better mixing performance compared with other settings.

## 7 Conclusion

We demonstrated the connection between HMC and slice sampling, introducing a new method for implementing a slice sampler via an augmented form of HMC. With few modifications to standard HMC, our MG-HMC can be seen as a drop-in replacement for any scenario where HMC and its variants apply, for example, Hamiltonian Variational Inference (HVI) [23]. We showed the theoretical advantages of our method over standard HMC, as well as numerical difficulties associated with it. Several future extensions can be explored to mitigate numerical issues, *e.g.*, performing MG-HMC on the Riemann manifold [5] so that step-sizes can be adaptively chosen, and using a high-order symplectic numerical method [24, 25] to reduce the discretization error introduced by the integrator.

## Footnotes

*For multidimensional cases, the Hamiltonian dynamics are semi-periodic, yet a similar conclusion still holds. Details are discussed in the Appendix.

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
