[Supplementary Material · nips_2016_suppl.pdf]

# Appendix for Towards Unifying Hamiltonian Monte Carlo and Slice Sampling

## A  Illustration of MG-SS with different monomial parameters $a$

Figure 4: MG-HMC and equivalent MG slice sampler. Red and blue dashed lines denote the conditionals $p(y_t|x_t)$ and $p(x_{t+1}|y_t)$, respectively.

The MG-SS for $0 < a < 1, a = 1$ and $a > 1$ are illustrated in Figure 4. When $0 < a < 1$, the conditional distribution $p(y_t|x_t)$ is skewed towards the current unnormalized density value $f(x_t)$. The conditional draw of $p(x_{t+1}|y_t)$ encourages taking samples with smaller density value (small moves), within the domain of the slice interval $\mathbb{X}$. On the other hand, when $a > 1$, draws of $y_t$ tend to take smaller values, while draws of $x_{t+1}$ encourage sampling from those with large density function values (large moves). Intuitively, setting $a$ to be small makes the auxiliary variable, $y_t$, stay close to $f(x_t)$, thus $f(x_{t+1})$ is close to $f(x_t)$. As a result, a larger $a$ seems more desirable.

## B  Monomial Gamma distribution

Several useful observations can be drawn from monomial Gamma distribution:

1) The mean and variance for it is 0 and $\frac{\Gamma(3a+1)}{3\Gamma(a+1)}m^{2a}$ , respectively.

2) Scaling. For any $\lambda > 0$,

$$Y \sim \text{MG}(a, m) \quad \Rightarrow \quad \lambda Y \sim \text{MG}(a, |\lambda|^{1/a}m)$$

3) As $a \to \infty$, the distribution, regardless of scale, becomes more heavy-tailed.

## C  Periodicity of Hamiltonian flow and higher dimensional HMC equivalents

First note that $\lim_{x\to\pm\infty} U(x) \to \infty$, since the integral $\int \exp(-U(x))$ is finite. From definition $\lim_{p\to\pm\infty} K(p) \to \infty$. Given above conditions, if the target distribution has one dimension, the Hamiltonian flow is periodic, and the Hamiltonian contour is closed [26].

In (5), $\int_{x_{min}}^{x(\tau)} f(z)^{a-1}dz \in [0, \int_{\mathbb{X}}[H - U(z)]^{a-1}dz]$. For one dimensional problems, the Hamiltonian dynamics described in (5) has a period $T \triangleq 2a \int_{\mathbb{X}}[H - U(z)]^{a-1}dz$. Since the $K(p)$ has symmetric form, the contour is symmetric along $p = 0$. In the second half of the period, the particle $x$ simply reverse the motion of the first half period.

However, if the dimensionality $D$ is higher than one, the periodicity assumption will almost never be true, the flow will typically be quasi-periodic as the periods of each 1D component would not exactly match. In those cases, the hamiltonian trajectory is a one-dimensional manifold in high-dimensional space. If uniformly sample a time $\tau$ from an interval with width much larger than $\prod_d T_d$, where $T_d$ is the period for $d$-th dimension, the hamiltonian trajectory will behave like a *dynamic billiard* [13]. With infinite evolutionary time, the trajectory will almost certainly cover each point in a hyper-rectangle $^\dagger$ $\mathbb{Y} = \{x : L_d \le x^{(d)} \le R_d, \text{ for all }, d \in \{1, \cdots, D\}\}$, which is one of the maximum

---

$^\dagger$suppose the $U(x)$ is decomposable over dimensions, if not the hyper-rectangle will become hyper-diamond in the high-dimensional space.

hyper-rectangles that lives within the slice interval $\mathbb{X} = \{x : U(x) \leq H\}$. For each dimension $d$, the boundary of the hyper-rectangle $L_d$ and $R_d$ are determined by the last sample point $\mathbf{x}_{t-1}$.

The Hamiltonian trajectory and corresponding slice interval are shown in Figure 5. As in univariate

Figure 5: 2D Hamiltonian trajectory and corresponding slice interval when $a = 0.5$ (left) and $a = 1$ (right).

cases, when $a = 0.5$, the Hamiltonian dynamic corresponds to a conditional density with less probability mass in the region with large $f(x)$. When $a = 1$, the Hamiltonian dynamic corresponds to a uniform density. However, in each of the case the density is constraint in the hyper-rectangle $\mathbb{Y}$. Thereby, in cases more than one dimension, even the MG-HMC with $a = 1$ is not exactly recovering standard slice sampling, but rather a *generalized slice sampler*. For simplicity, suppose the mass matrix is $m\mathbf{I}$, it can be shown that $K(p) \sim \text{Gamma}(D/a, m)$, thus, this generalized slice sampler has iterative procedure (11) as below (Figure 6)

$$p(y_t|x_t) \propto [\log f(x_t) - \log y_t]^{D/a-1}, s.t. 0 < y_t < f(x_t) \tag{10}$$

$$p(x_{t+1}|y_t) \propto [\log f(x_{t+1}) - \log y_t]^{a-1}, s.t. x_{t+1} \in \mathbb{Y} \tag{11}$$

Figure 6: 2D equivalent generalized slice sampler for MG-HMC $a = 1/2$. Red and blue dashed lines denote the conditionals $p(y_t|x_t)$ and $p(x_{t+1}|y_t)$, respectively. $x^{(1)}$ and $x^{(2)}$ denote the first and second dimension of target distribution

# D Connecting HMC with generalized kinetics and slice sampling

We show in 2 and 3 that the generalized kinetic form $K(p) = |p|^{1/2}, a > 0$ lead to MG-SS 2. In fact, for the generalized kinetic form with mass parameter and monomial parameters, $K(p) = |p|^{1/a}/m, a, m > 0$, the conclusion still holds. To see this, one may rederive equations (3)-(8) using this generalized kinetic energy.

The (3) becomes

$$U(x) + |\frac{dW(x)}{dx}|^{1/a}/m - p' = 0 \,. \tag{12}$$

Solving (12) for $W(x)$ gives

$$W(x) = \int_{x_{min}}^{x(\tau)} [mf(z)]^a dz + C \,, \qquad f(z) = \begin{cases} H(\cdot) - U(z), & z \in \mathbb{X} \\ 0, & z \notin \mathbb{X} \end{cases} \,, \tag{13}$$

Hence, from (2) we have

$$x' = m^a a \int_{x_{min}}^{x(\tau)} f(z)^{a-1} dz - \tau \,, \tag{14}$$

For the (14), Hamiltonian dynamics with generalized kinetics $K(p) = |p|^{1/a}/m, a, m > 0$ has period $2m^a a \int_{\mathbb{X}} [H(\cdot) - U(z)]^{1-a} dz$ and is symmetric along $p = 0$ (due to the symmetric form of the kinetic function). The system time, $\tau$, is uniformly sampled from a half-period of the Hamiltonian dynamics.

$$\tau \sim \text{Uniform}\left(-x', -x' + m^a a \int_{\mathbb{X}} [H(\cdot) - U(z)]^{a-1}\right)$$

.

The constant $m^a a$ does not matter because when we transform the problem using inverse CDF methods, this constant would diminish from the formulation due to the normalization. From the inverse CDF transform sampling method, uniformly sampling $\tau$ from half of a period and solving for $x^*$ from (5), are equivalent to directly sampling $x^*$ from the following density

$$p(x^*|H(\cdot)) \propto [H(\cdot) - U(x^*)]^{a-1} \,, \qquad \text{s.t.} \ H(\cdot) - U(x^*) \geq 0 \,. \tag{15}$$

Denote $y_t = e^{-H_t(\cdot)}$, by substituting $H_t(\cdot)$ with $y_t$ in (6), the conditional updates for this new sampling procedure can be rewritten as below, yielding the MG-SS with arbitrary momomial parameter $a > 0$, with conditional distributions defined as

Sampling a slice: $\quad p(y_t|x_t) = \dfrac{1}{\Gamma(a)f(x_t)}[\log f(x_t) - \log y_t]^{1-a} \,, \quad \text{s.t.} \ 0 < y_t < f(x_t) \,, \quad$ (16)

Conditional sampling: $\quad p(x_{t+1}|y_t) = \dfrac{1}{Z_2(y_t)}[\log f(x_{t+1}) - \log y_t]^{1-a} \,, \quad \text{s.t.} \ f(x_t) > y_t \,,$

(17)

Note that the mass parameter $m$ in generalized kinetic function will not influence the density (16) and (17)

# E Theoretical properties of MG sampler

## E.1 Convergence properties of MG-SS

Following [27] and [1], we show in below that the MG-SS is reversible and *Harris ergodic*. As a result, the chain is guaranteed to uniquely and asymptotically converge to the target distribution. Next, following standard slice sampler [27], we show that MG-SS is uniformly ergodic under the *Doeblin's conditions* [28] in Lemma 2[‡].

---

[‡]We hypothesize that this proposition holds for $0 < a < 1$, however we leave it for further investigation.

**Lemma 2** *(Uniform ergodicity) Suppose $f(\cdot)$ is bounded and has bounded support. If $a \geq 1$, the analytic MG-SS is uniformly ergodic.*

Geometric ergodicity is a less restrictive property compared to the uniformly ergodicity. In MG-SS, we hypothesize this requires $yZ_2'(y)$ to be non-increasing. Formal verification of these conditions is beyond the scope of this paper, thus is left as intersting future work.

### E.1.1 Invariance

**Theorem 3** *($\pi$-invariant) The Hamiltonian dynamics $S : (x, p) \to (x', p')$ with parameter $a$, is $\pi$-invariant.*

**Proof** First, the total Hamiltonian $H$ is perserved with Hamiltonian dynamics.

$$\frac{dH}{dt} = \frac{\delta H}{\delta p}\frac{dp}{dt} + \frac{\delta H}{\delta x}\frac{dx}{dt} = 0$$

Thus, for any $f(x, p)$

$$\mathbb{E}_{\pi(x,p)}f(x,p) = \int f(x,p)\frac{e^{-H(x,p)}}{Z_1}dxdp = \int f(x,p)\frac{e^{-H(S(x,p))}}{Z_1}|J_s|dxdp$$

By Liouville's theorem, $|J_s| = 1$. Therefore, $\mathbb{E}_{\pi(x,p)}f(x,p) = \mathbb{E}_{\pi(x,p)}f(S(x,p))$, the transformation $S$ is $\pi$-invariant. ∎

### E.1.2 Reversibility

**Proposition 4** *(Reversibility, detailed balance) For the transition kernel in eqn. (31). We have $\mathbb{E}_{\kappa_h^{-1}(x'|x)}x' = \mathbb{E}_{\kappa_h(x'|x)}x'.$*

**Proof** From the symmetric form of the joint distribution, we have $\kappa_1(x'|x)p(x) = \kappa_1^{-1}(x|x')p(x')$. Using induction we have $\kappa_h(x'|x)p(x) = \kappa_h^{-1}(x|x')p(x')$. Thus, $\mathbb{E}_{\kappa_h^{-1}(x'|x)}x' = \mathbb{E}_{\kappa_h(x'|x)}x'$ ∎

### E.1.3 Harris ergodicity

**Theorem 5** *(Harris ergodicity) The MG sampler with parameter $a$, is Harris ergodic with invariant distribution $p(x)$. $\kappa_h(\cdot, x)$ is the $h$-th transition kernel.*

$$\|\kappa_h(\cdot, x) - p(x)\|_{TV} \to 0, \text{ as } h \to \infty$$

*Further, $\|\kappa_h(\cdot, x) - p(x)\|$ is monotonically nonincreasing in h. (Meyn and Tweedie (1993), proposition 13.3.2)*

**Proof** Following Lemma 1 of Tan and Hobert (2008), it can be shown that MG sample is reversible, aperiodic and $\pi$-irreducible. The Harris recurrent property follows directly from Corollary 1 of Tierney (1994), which states that an $\pi$-irreducible Markov chain is Harris recurrent if for some $h$, $\kappa_h x, \cdot$ is absolutely continuous *w.r.t.* $p(x)$ for all $x \in \mathcal{X}$. ∎

Note that for MG-HMC, the Harris ergodicity cannot be directly extended from above conclusion, because 1) MG-HMC has fixed leap-frog step, 2) the Hamiltonian dynamics would not allow moving between contours with same energy. However, [29] showed that HMC is $\pi$-irreducible under the assumption that the potential energy has an upper bound. One can use the similar technique to show such conclusion holds for MG-HMC.

### E.1.4 Geometric ergodicity

Establishing such ergodicity for general cases requires demonstrating the *drift* and *minorisation* conditions [30]. [31] has showed that for any univariate log-concave density $f(\cdot)$, the resulting Markov chain associated with the slice sampler is geometrically ergodic, and the quantitative convergence

bounds are available. In fact, a necessary condition for any multivariate density being geometrically ergodic is that $y\mu'(y)$ is non-increasing with respect to $y$, where $\mu(y)$ is the Lebesgue measure of the slice interval $\{w : f(w) \geq y\}$, and the prime symbol denotes derivative *w.r.t.* $y$. In MG-SS, we hypothesize this requires $yZ_2'(y)$ to be non-increasing, However we leave the formal verification for future work.

### E.1.5 Uniform ergodicity

**Proposition 6** *(Uniformly ergodic) If $f(\cdot)$ is bounded and have bounded support, the analytic MG-SS with $a \geq 1$ is uniformly ergodic,* i.e. ,

$$\lim_{h\to\infty} \sup_{x_0 \in \mathcal{X}} \|\kappa_h(x_0, x) - p(x)\|_{TV} = 0.$$

**Proof** Following [27] and [1], without lossing generality, assume $f(\cdot) \in [0, 1]$ and the support for $f(\cdot)$ is [0,1]. A sufficient and necessary condition to demonstrate uniform ergodicity, is by *Doeblin's condition* [28]. The c.d.f. of transition kernel is given by,

$$\xi(v) \quad = \quad \Pr(f(x_{t+1}) > \eta \,| f(x_t) = v).$$

To establish Doeblin's condition, we will first show that $\xi(v)$ is decreasing in $v$ for $\forall \eta$, when $a > 1$. With some algebra one can obtain,

$$\xi(v) = \frac{1}{v\Gamma(a)} \int_0^{\eta \wedge v} \frac{Z_2(w) - Z_2(\eta)}{Z_2(w)} \cdot (\log v - \log w)^{a-1} dw$$

$$= \quad \frac{1}{v\Gamma(a)} \int_0^v \max\{K(w; \eta, v), 0\} dw,$$

Where $K(w; \eta, v) \triangleq \left(1 - \frac{Z_2(\eta)}{Z_2(w)}\right) \cdot (\log v - \log w)^{a-1}$. $Z_2(w) = \int_{f(x)>w}(\log f(x) - \log w)^{a-1} dx$. When $a > 1$, $Z_2(w)$ is decreasing in $w$. To see this, suppose $w_1 \geq w_2$,

$$Z_2(w_1) \quad \leq \quad \int_{f(x)>w_2} (\log x - \log w_1)^{a-1} dx$$

$$\leq \quad \int_{f(x)>w_2} (\log x - \log w_2)^{a-1} dx = Z_2(w_2).$$

Denote $\psi(w) = \left(1 - \frac{Z_2(\eta)}{Z_2(w)}\right)$. $\psi(w)$ is decreasing in $w$, and $\psi(w) > 0$ when $w \in (0, \eta)$ When $v \geq \eta$, we have,

$$\xi(v) \quad = \quad \frac{1}{v\Gamma(a)} \int_0^v K(w; \eta, v) dw.$$

Taking derivatives gives,

$$\xi'(v) \quad = \quad \frac{1}{v\Gamma(a)} \left[\frac{a-1}{v} \int_0^\eta \psi(w) \cdot (\log v - \log w)^{a-2} dw\right]$$

$$-\frac{1}{v^2\Gamma(a)} \int_0^\eta \psi(w) \cdot (\log v - \log w)^{a-1} dw$$

$$\triangleq \quad -\frac{1}{v^2\Gamma(a)} \int_0^\eta \psi(w) \cdot h(w) dw,$$

where we denote $h(w) = (\log v - \log w - a + 2)(\log v - \log w)^{a-2}$, it can be validated that $\int h(w) dw = 0$. Meanwhile, one can also validate that $\exists w_0 \in (0, v)$, where $h(w) > 0$ for $\forall w \in (0, w_0)$ and $h(w) < 0$ for $\forall w \in (w_0, v)$. Therefore,

$$\xi'(v) \quad = \quad -\frac{1}{v^2\Gamma(a)} \int_0^\eta [\psi(w) - \psi(w_0)] h(w) dw$$

$$= \quad -\frac{1}{v^2\Gamma(a)} \int_0^{w_0} [\psi(w) - \psi(w_0)] h(w) dw$$

$$-\frac{1}{v^2\Gamma(a)} \int_{w_0}^\eta [\psi(w) - \psi(w_0)] h(w) dw < 0.$$

The inequality follows because $\psi(w)$ is increasing in $w$. Likewise, one can obtain that When $v \leq \eta$, $\xi(v)$ can be written as,

$$\xi(v) \quad = \quad \frac{1}{v\Gamma(a)} \int_0^v K(w; \eta, v) dw$$

Thereby, similar to the case of $v \geq \eta$, we have,

$$\begin{aligned} \xi'(v) \quad &= \quad -\frac{1}{v^2\Gamma(a)} \int_0^v [\psi(w) - \psi(w_0)] h(w) dw \\ &= \quad -\frac{1}{v^2\Gamma(a)} \int_0^{w_0} [\psi(w) - \psi(w_0)] h(w) dw \\ &\quad -\frac{1}{v^2\Gamma(a)} \int_{w_0}^v [\psi(w) - \psi(w_0)] h(w) dw < 0. \end{aligned}$$

Thus, $\xi(v)$ is equally decreasing in $v$, where $v \in [0, f_0]$, $f_0 = \max(f(x))$. The upper and lower bound of $\xi(v)$ can be achieved by $\lim_{v \to 0} \xi(v)$ and $\lim_{v \to f_0} \xi(v)$, which are non-degenerate cdf for $\eta$. Thus one can establish uniform ergodicity via Doeblin's condition. The case $a = 1$ is standard slice sampling, and has been shown to be uniformly ergodic with bounded $f$ and have bounded support [27]. ∎

### E.2 Theoretical result about autocorrelation

#### E.2.1 Proof of $\rho_x(1) > 0$

We will first prove the Proposition 7, showing that $\rho_x(1) > 0$.

**Proposition 7** *The one-step autocorrelation, $\rho_x(1) \triangleq \rho(x_t, x_{t+1})$, is non-negative.*

From (7) and (8), and provided the conditional density $p(x_t|y_t)$ and $p(x_{t+1}|y_t)$ have the same form, we have

$$\mathbb{E}x_t x_{t+1} = \mathbb{E}_{p(y_t)}[\mathbb{E}_{p(x_t|y_t)} x_t \mathbb{E}_{p(x_{t+1}|y_t)} x_{t+1}] = \mathbb{E}_{p(y_t)}[\mathbb{E}_{p(x_{t+1}|y_t)} x_{t+1}]^2, \quad (18)$$

where $p(x|y)$ is the conditional distribution defined in (8). From (18), when $p(x)$ is *symmetric* at $x = c$ ($c$ being a constant) , $\mathbb{E}_{p(x_{t+1}|y_t)} x_t = \mathbb{E}x$, which gives $\rho_x(1) = 0$. From (18) and the Jensen's inequality, assuming the sampler has reached stationary period, we can obtain

$$[\mathbb{E}x]^2 = [\mathbb{E}_{p(y)} \mathbb{E}_{p(x|y)} x]^2 \leq \mathbb{E}x_t x_{t+1} \leq \mathbb{E}_{p(y)}[\mathbb{E}_{p(x|y)} x^2] = \mathbb{E}x^2$$

This indicate $0 \leq \rho_x(1) \leq 1$

#### E.2.2 Autocorrelation for $\{y_t\}_{t=1,2,\cdots}$

The analytic MG-SS performs sampling in an iterative manner, *i.e.*, $x_t \to y_t \to x_{t+1} \to y_{t+1} \cdots$. To gain insights about the limiting behavior of $\{x_t\}_{t=1,\ldots}$, when $a$ goes to infinity, we first consider the Markov Chain of $\{y_t\}_{t=1,\ldots}$, which can be analytically calculated regardless of the form of $U(x)$. Particularly, we will show that $\lim_{a \to \infty} \rho(y_t, y_{t+1}) = 0$. Also, that $\rho(y_t, y_{t+h})$ is a non-negative decreasing function of the time lag in discrete steps $h = 1, 2, \ldots$.

We start by finding the autocorrelation $\rho(y_t, y_{t+1})$. First consider compute $\rho(H_t, H_{t+1})$, where $H = -\log y$.

$$\mathbb{E}H_t H_{t+1} = \mathbb{E}_{p(x_t)}\mathbb{E}_{p(H_t|x_t)}H_t\mathbb{E}_{p(H_{t+1}|x_t)}H_{t+1}$$

$$= \mathbb{E}_{p(x_t)}\left[\frac{\Gamma(a)(a+U(x_t))e^{-U(x_t)}}{\Gamma(a)e^{-U(x_t)}}\right]^2 = \mathbb{E}_{p(x)}[a+U(x)]^2$$

$$\mathbb{E}H = \frac{1}{\Gamma(a)Z_1}\int HB(H)e^{-H}dH$$

$$= \frac{1}{\Gamma(a)Z_1}\int \Gamma(a)[a+U(x_t)]e^{-U(x_t)}dx_t = \mathbb{E}_{p(x)}[a+U(x)]$$

$$\text{Var}(H) = \mathbb{E}H^2 - (\mathbb{E}H)^2 = \mathbb{E}_{p(x)}(a+[a+U(x)]^2) - (\mathbb{E}_{p(x)}[a+U(x)])^2$$

$$\rho(H_t, H_{t+1}) = \frac{\mathbb{E}H_t H_{t+1} - (\mathbb{E}H)^2}{\text{Var}(H)} = \frac{\text{Var}_{p(x)}U(x)}{a + \text{Var}_{p(x)}U(x)}$$

Since the mapping from $y(H) = \exp(-H)$ is bijective, after some algebra we can obtain $\lim_{a\to\infty}\rho(y_t, y_{t+1}) = 0$. Similarly, for two-step autocorrelation one can obtain

$$\rho(H_t, H_{t+2}) = \frac{\mathbb{E}H_t H_{t+2} - (\mathbb{E}H)^2}{\text{Var}(H)} = \frac{\mathbb{E}_{p(H)}[\mathbb{E}_{p(x|H)}U(x)]^2 - (\mathbb{E}_{p(x)}U(x))^2}{a + \text{Var}_{p(x)}U(x)}$$

Thereby, one can obtain

$$0 \leq \rho(H_t, H_{t+2}) \leq \rho(H_t, H_{t+1}) \tag{19}$$

It not hard to further obtain that $\rho(H_t, H_{t+h})$ is a non-negative decreasing function of step $h$. After some algebra, the $\rho(y_t, y_{t+h})$ also has this monotonicity w.r.t. $h$. The property of $y$ as part of the markov chain gives some intuition on the behavior of $x$.

### E.2.3 Proof of Distillation theory

In order to describe the limiting behavior of $\rho(x_t, x_{t+1})$, We first establish following Lemma

**Lemma 8** *Define $\hat{B}(0, \epsilon)$ to be the $d-$dimensional ball around zero with a radius of $\epsilon$ and $p_0, k$ be some positive constants, then for sufficient large $a$ and any $\epsilon = \sqrt{p_0 k}\frac{\log a}{\sqrt{a+1}}$, we have*

$$g(a, d, \epsilon) = \int_{\hat{B}(0,\epsilon)}\left(1 - \frac{\|t\|^2}{2p_0}\right)^a dt \geq \frac{d\pi^{d/2}}{\Gamma(d/2+1)}\frac{(d-2)!!p_0^{d/2}}{k(a+1)^{d/2}\log a}\left\{1 - \frac{(d+1)(k\log a)^{\frac{d+1}{2}}}{2a^{k/2}}\right\}.$$

**Proof** We will prove the result first for $d = 1$ and $d = 2$ and then extends the result to $d \geq 3$ by mathematical induction.

For $d = 1$, it is hard to directly evaluate $g(a, 1, \epsilon)$, but we notice that

$$g(a, 1, \epsilon) \geq \frac{2}{\epsilon}\int_0^\epsilon t\left(1 - \frac{t^2}{2p_0}\right)^a dt = \frac{2p_0}{\epsilon(a+1)} - \frac{2p_0}{\epsilon(a+1)}\left(1 - \frac{\epsilon^2}{2p_0}\right)^a.$$

Taking $\epsilon = \sqrt{p_0 k}\frac{\log(a+1)}{\sqrt{a+1}}$ for any $k > 0$, we have

$$g(a, 1, \epsilon) \geq \frac{2p_0^{1/2}}{k(a+1)^{1/2}\log a}\left\{1 - \left(1 - \frac{\epsilon^2}{2p_0}\right)^a\right\} \geq \frac{\pi^{1/2}}{\Gamma(\frac{3}{2})}\frac{p_0^{1/2}}{k(a+1)^{1/2}\log a}\left\{1 - \frac{1}{a^{k/2}}\right\}.$$

For $d = 2$, we transform the integral to polar coordinate transformation and obtain that

$$g(a, 2, \epsilon) = \pi\int_0^\epsilon r\left(1 - \frac{r^2}{2p_0}\right)^a dr = \frac{\pi p_0}{a+1} - \frac{\pi p_0}{a+1}\left(1 - \frac{\epsilon^2}{2p_0}\right)^a.$$

where $r$ is an unknown positive constant. With the same $\epsilon$, we have

$$g(a, 2, \epsilon) \geq \frac{2\pi}{\Gamma(2)}\frac{p_0}{k(a+1)\log a}\left\{1 - \frac{1}{a^{k/2}}\right\}.$$

Consequently, we would generally guess that for any $d \geq 3$, we have

$$g(a, d, \epsilon) \geq \frac{d\pi^{d/2}}{\Gamma(d/2 + 1)} \frac{(d-2)!! p_0^{d/2}}{k(a+1)^{d/2} \log a} \left\{ 1 - \frac{(d+1)(k \log a)^{\frac{d+1}{2}}}{2a^{k/2}} \right\}. \tag{20}$$

We use mathematical induction to prove the this inequality. It has already been verified for $d = 1$ and $d = 2$. For $d \geq 3$, using polar coordinate, the general $g(a, d, \epsilon)$ can be written as

$$g(a, d, \epsilon) = \frac{d\pi^{d/2}}{\Gamma(d/2 + 1)} \int_0^\epsilon r^{d-1} \left( 1 - \frac{r^2}{2p_0} \right)^a dr,$$

Using integration by parts, we have

$$\int_0^\epsilon r^{d-1} \left( 1 - \frac{r^2}{2p_0} \right)^a dr = \frac{1}{d} r^d \left( 1 - \frac{r^2}{2p_0} \right)^a \Big|_0^\epsilon + \frac{a}{dp_0} \int_0^\epsilon r^{d+1} \left( 1 - \frac{r^2}{2p_0} \right)^{a-1} dr$$

$$= \frac{\epsilon^d}{d} \left( 1 - \frac{\epsilon^2}{2p_0} \right)^a + \frac{a}{dp_0} \int_0^\epsilon r^{d+1} \left( 1 - \frac{r^2}{2p_0} \right)^{a-1} dr.$$

This implies that

$$g(a, d, \epsilon) = \frac{(d-2)p_0}{a+1} g(d-2, a+1, \epsilon) - \frac{p_0 \epsilon^{d-2}}{a+1} \left( 1 - \frac{\epsilon^2}{2p_0} \right)^{a+1}.$$

Using the induction, we have

$$g(a, d, \epsilon) \geq \frac{d\pi^{\frac{d}{2}}}{\Gamma(\frac{d}{2} + 1)} \frac{(d-2)!! p_0^{d/2}}{k(a+1)^{d/2} \log a} \left\{ 1 - \frac{(d-1)(k \log(a+1))^{\frac{d-1}{2}}}{2(a+1)^{k/2}} \right\} - \frac{d\pi^{\frac{d}{2}}}{\Gamma(\frac{d}{2} + 1)} \frac{p_0 \epsilon^{d-2}}{a+1} \left( 1 - \frac{\epsilon^2}{2p_0} \right)^{a+1}$$

Since $\epsilon = \sqrt{p_0 k} \frac{\log a}{\sqrt{a+1}}$, the second term can be upper bounded as

$$\frac{p_0 \epsilon^{d-2}}{a+1} \left( 1 - \frac{\epsilon^2}{2p_0} \right)^{a+1} \leq \frac{p_0^{d/2}}{k(a+1)^{d/2} \log a} \cdot \frac{(k \log a)^{d/2}}{(a+1)^{k/2}}$$

Thus, we have

$$g(a, d, \epsilon) \geq \frac{d\pi^{\frac{d}{2}}}{\Gamma(\frac{d}{2} + 1)} \frac{(d-2)!! p_0^{d/2}}{k(a+1)^{d/2} \log a} \left\{ 1 - \frac{(d-1)(k \log(a+1))^{\frac{d-1}{2}} + (k \log(a+1))^{\frac{d}{2}}}{2(a+1)^{k/2}} \right\}$$

$$\geq \frac{d\pi^{\frac{d}{2}}}{\Gamma(\frac{d}{2} + 1)} \frac{(d-2)!! p_0^{d/2}}{k(a+1)^{d/2} \log a} \left\{ 1 - \frac{(d+1)(k \log(a+1))^{\frac{d+1}{2}}}{2(a+1)^{k/2}} \right\},$$

which completes the proof. ∎

From Lemma 8, we give the Lemma 9 as below, in order to describe the limiting behavior of conditional density $p(x|H)$

**Lemma 9** *(Distillation) Let $p(x)$ be a non-negative integrable function defined on $x \in \mathcal{D}$, where $\mathcal{D} \subseteq \mathcal{R}^d$. Assume $p(x)$ is thrice differentiable with the third-order derivative being bounded. Define $\mathcal{M} = \{x : x = argmax_x(p(x))\}$ to be the collection of all maximum point(s) of $p(x)$. We assume $p(x)$ is locally concave on $\mathcal{M}$, i.e., $\nabla^2 p(x)$ is negative definite for any $x \in \mathcal{M}$. Define a measure on $\mathcal{M}$ as*

$$\mu(x) \propto -\nabla^2 p(x), \quad \forall x \in \mathcal{M},$$

*then for any $r > 0$ and sufficiently large $a$, we have the following result*

$$\left\| \frac{\int x p(x)^a dx}{\int p(x)^a dx} - \mathbb{E}_\mu X \right\| = \mathcal{O}\left\{ p_0^{1/2} \frac{\log a}{\sqrt{a+1}} + \left( \frac{\int p(x)}{p_0^{d/2+1}} + \frac{\int \|x\|_1 p(x)}{p_0^{d/2+1}} \right) \frac{\log a}{(a+1)^r} \right\},$$

*where $p_0 = \max p(x)$.*

Intuitively, Lemma 9 states that when $a \to \infty$, the limiting expectation of distribution $p(x; a)$ will be distilled to the expectation over the domain that maximizes $g(x)$. Assume that our target density function contains no singular points, and $U(x) = -\log f(x)$ has minimum value. From Lemma 9, for any feasible $H$, when $a \to \infty$, we can obtain $\mathbb{E}_{p(x|H)} x = \mathbb{E}_{u(x)} x, x \in \mathcal{M}$, which are the expectation of the maximum point(s) of $g(x) \triangleq H - U(x)$ (or, minimum point(s) of $U(x)$) that do not depend on $H$. Based on this result, one can establish a Theorem 1 describing the limiting behavior of $\rho_x(1)$,

**Proof** We consider the case where $\mathcal{M}$ is a finite set, which is the most common scenario. The proof for $\mathcal{M}$ being positive measure follows similarly and will be ignored here.

Let $\mathcal{E}(x, \epsilon, \nu, \rho) = \{ y : (y - x)^\top (\Sigma_x + 2\nu\rho I)(y - x) \le \epsilon^2 \}$ denote the elliptical ball around point $x$ and $\$\mathcal{E}(\mathcal{M}, \epsilon, \nu) = \bigcup_{x_i \in \mathcal{M}} \mathcal{E}(x_i, \epsilon, \nu, \rho_i)$, where the positive definite matrix $\Sigma_x = -\nabla^2 p(x)$ if $x$ is the maximum point and any constants $\nu, \rho, \epsilon > 0$. The maximum radius of the elliptical ball $\rho_{\max}(\nu) = (\tilde{\lambda}_{\min} + 2\nu\rho)^{-1/2}\epsilon$, where $\tilde{\lambda}$ is the eigen value of $\Sigma_x$. By solving the equation $2\nu\rho^3 + \tilde{\lambda}_{\min}\rho^2 - \epsilon^2 = 0$ w.r.t. $\rho > 0$, we can obtain a stable or determined $\rho_{\max}$ and we will use this stable value later. Similarly, we can compute $\rho_{\min}(\nu) = (\tilde{\lambda}_{\max} + 2\nu\rho)^{-1/2}\epsilon$.

By the maximality of $\mathcal{M}$ and the smoothness of $A(H)/B(H)$ ($A(H)$ and $B(H)$ are defined in Section F), there must exist some $\epsilon_0$ such that for any $\epsilon_1, \epsilon_2 \le \epsilon_0$, we have

$$\mathcal{E}(x_1, \epsilon_1, \nu, \rho_1) \cap \mathcal{E}(x_2, \epsilon_2, \nu, \rho_2) = \emptyset, \quad \forall x_1, x_2 \in \mathcal{M},$$

where $A^c = \mathcal{D} \setminus A$. In addition, since we assume $p(x)$ is locally concave at all $x_i \in \mathcal{M}$ and the second-order derivative of $p(x)$ is continuous, we can quantify the local behavior of $p(x)$ at each point in $\mathcal{M}$ by using the Taylor expansion,

$$p(x) - p(x_i) = \nabla p(x_i)(x - x_i) + \frac{1}{2}(x - x_i)^T \nabla^2 p(x_i)(x - x_i) + \mathcal{O}(\|x - x_i\|^3),$$

where the tail form is due to the existence of the third-order derivative. By definition we have $\nabla p(x_i) = 0$. The local concavity and smoothness ensures that $\nabla^2 p(x_i)$ is negative definite and the largest and smallest eigenvalue can be controlled by some constants, i.e., there exists some constants $L \ge l > 0$, such that

$$-L \le \lambda_{\min}(\nabla^2 p(x_i)) \le \lambda_{\max}(\nabla^2 p(x_i)) \le -l, \quad \forall x_i \in \mathcal{M}.$$

Therefore, when $\epsilon$ is sufficiently small, we can obtain $\rho_{\max}$ is small as well, and expect $p(x)$ can be well approximated by some local quadratic function. More precisely, defining $p_0 = p(x_i) = \max_{x \in \mathcal{D}} p(x)$, there exists some $\nu > 0, \epsilon'_0 > 0, \epsilon'_0 < \epsilon_0$ such that,

$$p(x) - p_0 \ge \frac{1}{2}(x - x_i)^T \nabla^2 p(x_i)(x - x_i) - \nu\|x - x_i\|^3 \ge -\frac{L}{2}\|x - x_i\|^2 - \nu\|x - x_i\|^3 \quad (21)$$

and

$$p_0 - p(x) \ge -\frac{1}{2}(x - x_i)^T \nabla^2 p(x_i)(x - x_i) - \nu\|x - x_i\|^3 \ge \frac{l}{2}\|x - x_i\|^2 - \nu\|x - x_i\|^3, \quad (22)$$

when $x \in \mathcal{E}(x_i, \epsilon'_0, \nu, \rho_i)$ for any $x_i \in \mathcal{M}$. The constants chosen are not the tightest, but adequate for the proof.

Now for any $\epsilon > 0$, we will partition the space into $\mathcal{E}(\mathcal{M}, \epsilon, \nu)$ and $\mathcal{D} \setminus \mathcal{E}(\mathcal{M}, \epsilon, \nu)$ (short noted as $\mathcal{E}(\mathcal{M}, \epsilon, \nu)^c$), and the target can then be written as

$$\frac{\int x p(x)^a dx}{\int p(x)^a dx} = \frac{\int x \left(\frac{p(x)}{p_0}\right)^a dx}{\int \left(\frac{p(x)}{p_0}\right)^a dx} = \frac{\int_{\mathcal{E}(\mathcal{M}, \epsilon, \nu)} x \left(\frac{p(x)}{p_0}\right)^a dx + \int_{\mathcal{E}(\mathcal{M}, \epsilon, \nu)^c} x \left(\frac{p(x)}{p_0}\right)^a dx}{\int_{\mathcal{E}(\mathcal{M}, \epsilon, \nu)} \left(\frac{p(x)}{p_0}\right)^a dx + \int_{\mathcal{E}(\mathcal{M}, \epsilon, \nu)^c} \left(\frac{p(x)}{p_0}\right)^a dx} = \frac{F_1(a) + F_2(a)}{G_1(a) + G_2(a)}.$$

The proof will be done by bounding the four terms and will be divided into two parts.

**1. Bounding $F_2$ and $G_2$** We first look at $F_2$ and $G_2$. When $\epsilon$ is small compared to $\epsilon'_0$, $\max_{\hat{B}(\mathcal{M}, \epsilon)^c} p(x)$ will be achieved at some point inside $\mathcal{E}(\mathcal{M}, \epsilon'_0, \nu)$. This is easy to show. Since $p_0 > \max_{\mathcal{E}(\mathcal{M}, \epsilon'_0, \nu)^c} p(x)$, we can always find an $\epsilon''_0 < \epsilon'_0$, such that $\min_{\mathcal{E}(\mathcal{M}, \epsilon''_0, \nu)} p(x) > \max_{\mathcal{E}(\mathcal{M}, \epsilon_0, \nu)^c} p(x)$. Now for any $\epsilon < \epsilon''_0$, $\max_{\mathcal{E}(\mathcal{M}, \epsilon, \nu)^c} p(x)$ must be achieved within $\mathcal{E}(\mathcal{M}, \epsilon'_0, \nu)$.

We assume $\epsilon_0' = \epsilon_0''$ in the proof just for simplicity. The above argument along with (22) suggests that for sufficiently small $\epsilon$, it holds that

$$p_0 - \max_{\mathcal{E}(\mathcal{M},\epsilon,\nu)^c} p(x) \geq \frac{l}{3} \min_{\mathcal{M}} \rho_{\min}^2.$$

Therefore, for $a \geq 1$, we can obtain an explicit decaying rate on $F_2(a)$ and $G(a)$ as

$$|F_2(a)| = \left| \int_{\mathcal{E}(\mathcal{M},\epsilon,\nu)^c} x \left( \frac{p(x)}{p_0} \right)^{a-1} \frac{p(x)}{p_0} dx \right| \leq C_1 \left( 1 - \frac{l \min_{\mathcal{M}} \rho_{\min}^2}{3p_0} \right)^{a-1},$$

where $C_1 \leq \frac{\int \|x\|_1 p(x)}{p_0}$ is a absolute constant only depending on $p(x)$. Similarly, we have for $G_2(a)$ that

$$G_2(a) = \int_{\mathcal{E}(\mathcal{M},\epsilon,\nu)^c} \left( \frac{p(x)}{p_0} \right)^{a-1} \frac{p(x)}{p_0} dx \leq C_2 \left( 1 - \frac{l \min_{\mathcal{M}} \rho_{\min}^2}{3p_0} \right)^{a-1},$$

where $C_2 \leq \int p(x)/p_0$ is an absolute constant depending on $p(x)$.

**2. Bounding $F_1$ and $G_1$**    Next, we consider $F_1(a)$ and $G_1(a)$. Notice that we have

$$F_1(a) = \sum_{x_i \in \mathcal{M}} \int_{\mathcal{E}(x_i,\epsilon,\nu,\rho_i)} x \left( \frac{p(x)}{p_0} \right)^a dx = \sum_{x_i \in \mathcal{M}} x_i \int_{\mathcal{E}(x_i,\epsilon,\nu,\rho_i)} \left( \frac{p(x)}{p_0} \right)^a dx$$
$$+ \sum_{x_i \in \mathcal{M}} \int_{\mathcal{E}(x_i,\epsilon,\nu,\rho_i)} (x - x_i) \left( \frac{p(x)}{p_0} \right)^a dx.$$

In addition, we notice by Cauchy-Schwarz inequality that

$$\left| \int_{\mathcal{E}(x_i,\epsilon,\nu,\rho_i)} (x - x_i) \left( \frac{p(x)}{p_0} \right)^a dx \right| \leq \rho_{\max} \int_{\mathcal{E}(x_i,\epsilon,\nu,\rho_i)} \left( \frac{p(x)}{p_0} \right)^a dx,$$

Therefore, the key to bound $F_1$ and $G_1$ is to bound the integral $\int_{\mathcal{E}(x_i,\epsilon,\nu,\rho_i)} \left( \frac{p(x)}{p_0} \right)^a dx$, for which we have

$$\int_{\mathcal{E}(x_i,\epsilon,\nu,\rho_i)} \left( \frac{p(x)}{p_0} \right)^a dx \geq \int_{\mathcal{E}(x_i,\epsilon,\nu,\rho_i)} \left( 1 - \frac{-(x-x_i)^T \nabla^2 p(x_i)(x-x_i) + 2\nu\|x-x_i\|^3}{2p_0} \right)^a dx.$$

If we transform $x - x_i$ to a new variable $t$, we can obtain a simpler form as

$$\int_{\mathcal{E}(x_i,\epsilon,\nu,\rho_i)} \left( \frac{p(x)}{p_0} \right)^a dx \geq \int_{\mathcal{E}(x_i,\epsilon,\nu,\rho_i)} \left( 1 - \frac{t^T(\Sigma_i + 2\nu\epsilon)t}{2p_0} \right)^a dt.$$

This form can be further simplified by doing transformation $t' = (\Sigma_i + 2\nu\rho_i I)^{1/2} t$ on the right, the key point is to set $\rho_i$ as the solution of equation $2\nu\rho^3 + \tilde{\lambda}_{\min}(\Sigma_i)\rho^2 - \epsilon^2 = 0$, i.e. fix point $\rho_{\max}$. We have

$$\int_{\mathcal{E}(x_i,\epsilon,\nu,\rho_i)} \left( \frac{p(x)}{p_0} \right)^a dx \geq |\Sigma_i + 2\nu\rho_{\max}|^{-1/2} \int_{\hat{B}(0,\epsilon)} \left( 1 - \frac{\|t\|^2}{2p_0} \right)^a dt. \qquad (23)$$

(23) indicates that $\int_{\mathcal{E}(x_i,\epsilon,\nu,\rho_i)} \left( \frac{p(x)}{p_0} \right)^a dx$ will be lower bounded $\int_{\hat{B}(0,\epsilon)} \left( 1 - \frac{\|t\|^2}{2p_0} \right)^a dt$. Thus, the remaining task is to compare this quantity with $F_2$ and $G_2$. For a brief summary on this part, we define

$$w_i = \frac{\int_{\mathcal{E}(x_i,\epsilon,\nu,\rho_i)} \left( \frac{p(x)}{p_0} \right)^a dx}{\int_{\hat{B}(0,\epsilon)} \left( 1 - \frac{\|t\|^2}{2p_0} \right)^a dt} \quad \text{and} \quad g(a,d,\epsilon) = \int_{\hat{B}(0,\epsilon)} \left( 1 - \frac{\|t\|^2}{2p_0} \right)^a dt$$

we have $w_i \geq |\Sigma_i + 2\nu\rho_{\max}|^{-1/2}$.

Similarly, consider the elliptical ball $\mathcal{E}(x_i, \epsilon, -\nu, \rho_i)$ and fixed point $\rho_{\min}$ for equation $\rho = (\tilde{\lambda}_{\max}(\Sigma_i) - 2\nu\rho)^{-1/2}\epsilon$, we can obtain

$$\int_{\mathcal{E}(x_i,\epsilon,-\nu,\rho_i)} \left(\frac{p(x)}{p_0}\right)^a dx \le |\Sigma_i - 2\nu\rho_{\min}|^{-1/2} \int_{\hat{B}(0,\epsilon)} \left(1 - \frac{\|t\|^2}{2p_0}\right)^a dt. \tag{24}$$

Notice we can pick up a very small $\epsilon'$ (and the resulted $\nu$) for (24) s.t. $\rho'_{\min}$ is bigger than the $\rho_{\max}$ in (23). This is possible since the equation (24) is actually derived by defining new elliptical ball from the beginning of this proof, i.e.

$$\int_{\mathcal{E}(x_i,\epsilon,\nu,\rho_i)} \left(\frac{p(x)}{p_0}\right)^a dx \le \int_{\mathcal{E}(x_i,\epsilon',-\nu',\rho'_i)} \left(\frac{p(x)}{p_0}\right)^a dx \le |\Sigma_i - 2\nu\rho_{\min}|^{-1/2} \int_{\hat{B}(0,\epsilon')} \left(1 - \frac{\|t\|^2}{2p_0}\right)^a dt. \tag{25}$$

and then we have $w_i \le |\Sigma_i - 2\nu'\rho'_{\min}|^{-1/2}$, i.e.

$$|\Sigma_i + 2\nu\rho_{\max}|^{-1/2} \le w_i \le |\Sigma_i - 2\nu'\rho'_{\min}|^{-1/2}. \tag{26}$$

Additionally,

$$\frac{F_1(a)}{g(a,d,\epsilon)} = \sum_{x_i \in \mathcal{M}} w_i x_i + \mathcal{O}(\epsilon|\mathcal{M}|), \qquad \frac{G_1(a)}{g(a,d,\epsilon)} = \sum_{x_i \in \mathcal{M}} w_i$$

**3. Synthesizing the results** The value of $g(a,d,\epsilon)$ has been evaluated in Lemma 8. When $\epsilon$ is chosen to be $\epsilon = \sqrt{p_0 k} \frac{\log a}{\sqrt{a+1}}$, we have

$$g(a,d,\epsilon) \ge \frac{C_3 p_0^{d/2}}{k(a+1)^{\frac{d}{2}} \log a},$$

where $C_3$ is an absolute constant depends only on $d$. On the other hand, we have

$$\frac{G_2(a)}{g(a,d,\epsilon)} \le \frac{C_2 k(a+1)^{\frac{d}{2}} \log a}{C_3 p_0^{d/2}} \left(1 - \frac{l\epsilon^2}{3p_0}\right)^{a-1} \le \frac{C_2 k(a+1)^{\frac{d}{2}} \log a}{C_3 p_0^{d/2}} \frac{1}{2(a+1)^{\frac{kl}{3}}}$$

$$= \frac{C_2 k \log a}{2C_3 p_0^{d/2}}(a+1)^{\frac{d}{2} - \frac{kl}{3}} = \mathcal{O}\left(C_2 \frac{\log a}{(a+1)^r}\right),$$

as long as we choose $k \ge \frac{3(d+2r)}{2l}$ for some $r > 0$. Similar result holds for $F_2(a)$ as well. Therefore, we have

$$\frac{F_1(a) + F_2(a)}{F_2(a) + G_2(a)} = \frac{\frac{F_1(a)}{g(a,d,\epsilon)} + \frac{F_2(a)}{g(a,d,\epsilon)}}{\frac{G_1(a)}{g(a,d,\epsilon)} + \frac{G_2(a)}{g(a,d,\epsilon)}} = \frac{\sum_{x_i \in \mathcal{M}} w_i x_i + \mathcal{O}\left(\frac{\int \|x\|_1 p(x)}{p_0^{d/2+1}} \frac{\log a}{(a+1)^r}\right)}{\sum_{x_i \in \mathcal{M}} w_i + \mathcal{O}(\epsilon|\mathcal{M}|) + \mathcal{O}\left(\frac{\int p(x)}{p_0^{d/2+1}} \frac{\log a}{(a+1)^r}\right)}.$$

The relationship (26) entails that

$$w_i = |\Sigma_i|^{-1} + \mathcal{O}(\epsilon),$$

and placing the value of $\epsilon$ into the equation, we finally have

$$\frac{F_1(a) + F_2(a)}{F_2(a) + G_2(a)} = \frac{\sum_{x_i \in \mathcal{M}} |\Sigma_i|^{-1} x_i + \mathcal{O}\left(\frac{\int \|x\|_1 p(x)}{p_0^{d/2+1}} \frac{\log a}{(a+1)^r}\right)}{\sum_{x_i \in \mathcal{M}} |\Sigma_i|^{-1} + \mathcal{O}\left(p_0^{1/2} \frac{\log a}{\sqrt{a+1}}\right) + \mathcal{O}\left(\frac{\int p(x)}{p_0^{d/2+1}} \frac{\log a}{(a+1)^r}\right)},$$

given the points in $\mathcal{M}$ are bounded, which is true in this case. It then follows naturally that

$$\left\| \frac{\int x p(x)^a dx}{\int p(x)^a dx} - \frac{\sum_{x_i \in \mathcal{M}} |\Sigma_i|^{-1} x_i}{\sum_{x_i \in \mathcal{M}} |\Sigma_i|^{-1}} \right\| = \mathcal{O}\left\{ p_0^{1/2} \frac{\log a}{\sqrt{a+1}} + \left(\frac{\int p(x)}{p_0^{d/2+1}} + \frac{\int \|x\|_1 p(x)}{p_0^{d/2+1}}\right) \frac{\log a}{(a+1)^r} \right\}.$$

This completes the whole proof. ∎

### E.2.4  Proof of Theorem 1

Now it's ready to prove the Theorem 1, i.e. $\lim_{a\to\infty}\rho_x(1)=0$. First, to gain some intuitions, we see from Section F that for the exponential family class of model (with potential function $U(x)=x^\omega$), the $\mathbb{E}_{p(x|H)}x=\frac{A(H)}{B(H)}$ is at order $\mathcal{O}(H^{1/\omega}a^{-1/\omega})$ (using $A(H)$ and $B(H)$ as defined in Section F, i.e. $A(H)=\int_{U(x)\le H}x\cdot g(x,H)^{a-1}dx$, $B(H)=\int_{U(x)\le H}g(x,H)^{a-1}dx$, see Section F for more details). In fact, for all of our verified cases that can be analytically derived in Section F, $\mathbb{E}_{p(x|H)}x$ can be expressed in $\mathcal{O}(H^r a^{-r})$, where $r$ is a positive constant. Generally, if $\frac{A(H)}{B(H)}$ can be expressed in given form, one can verify below proposition, which leads to $\lim_{a\to\infty}\rho_x(1)=0$.

**Proposition 10** *If* $\mathbb{E}_{p(x|H)}x=\frac{A(H)}{B(H)}$ *can be written as* $\mathbb{E}_{p(x)}x\left(\sum_{r=0}^{\infty}\frac{s_r f_r(H)}{g_r(a)}\right)$, *where* $\lim_{H\to\infty}f_r(H)/H^r=1$, $\lim_{H\to\infty}g_r(a)/a^r=1$ *and* $\sum_{r=0}^{\infty}s_r=1$, *one can obtain* , $\lim_{a\to\infty}\rho_x(1)=0$.

**Proof** It can be shown that

$$\lim_{a\to\infty}\mathbb{E}x_t x_{t+1}=\lim_{a\to\infty}\frac{1}{\Gamma(a)Z_1}\int\frac{A(H)^2}{B(H)}e^{-H}dH$$

$$=\frac{\mathbb{E}_p x}{Z_1}\lim_{a\to\infty}\frac{1}{\Gamma(a)}\int\left(\sum_{r=0}^{\infty}\frac{s_r f_r(H)}{g_r(a)}\right)A(H)e^{-H}dH \tag{27}$$

$$=\frac{\mathbb{E}_p x}{Z_1}\sum_{r=0}^{\infty}s_r\lim_{a\to\infty}\frac{1}{\Gamma(a)}\int\frac{f_r(H)}{g_r(a)}A(H)e^{-H}dH \tag{28}$$

The (28) follows by Fubini's theorem. Let $K=H-U(x)$, for any $r\ge 0$, we have,

$$\lim_{a\to\infty}\frac{1}{\Gamma(a)}\int\frac{f_r(H)}{g_r(a)}A(H)e^{-H}dH$$

$$=\lim_{a\to\infty}\int\frac{xe^{-U(x)}}{\Gamma(a)}\int\frac{K^r+\mathcal{O}(K^{r-1}U(x)^r)}{a^r+\mathcal{O}(a^{r-1})}K^{a-1}e^{-K}dKdx$$

$$=Z_1\mathbb{E}_p x\cdot\lim_{a\to\infty}\frac{\Gamma(a+r)+\mathcal{O}(U(x)^r)\cdot\Gamma(a+r-1)}{\Gamma(a)(a^r+\mathcal{O}(a^{r-1}))}=Z_1\mathbb{E}_p x$$

Taking together with (28), we have

$$\lim_{a\to\infty}\mathbb{E}x_t x_{t+1}=(\mathbb{E}_{p(x)}x)^2$$

Thus, $\lim_{a\to\infty}\rho_x(1)=0$ ∎

This establish a sufficient condition for limitation of $\rho_x(1)$ goes to zero. This characterizes the order of $\mathbb{E}_{p(x|H)}x$.

We further provide a proof of Theorem 1 for more general cases in univariate setup, based on Lemma 9 (distillation)

**Theorem 11** *For a univariate target distribution, i.e. $\exp[-U(x)]$ has finite integral over $\mathbb{R}$, if $U(x)$ is thrice differentiable with bounded third-order derivative, the one-step autocorrelation of the MG-SS parameterized by $a$, asymptotically approaches zero as $a\to\infty$, i.e., $\lim_{a\to 0}\rho_x(1)=0$.*

**Proof**

Let $p(H)\triangleq\frac{1}{\Gamma(a)Z_1}B(H)e^{-H}$. From Lemma 9, one can obtain that $\lim_{a\to\infty}\frac{A(H)}{B(H)}=C_0$ where $C_0$ is a constant that is independent with $H$, where the convergence ratio is characterized by $\mathcal{O}\left\{\frac{\log a}{\sqrt{a+1}}+\frac{\log a}{(a+1)^r}\right\}$ where $r$ is the unknown constant in Lemma 8. As a result,

$$\lim_{a\to\infty}\left(\frac{A(H)}{B(H)}-\mathbb{E}_{p(H)}\frac{A(H)}{B(H)}\right)=0,\text{ or, }\lim_{a\to\infty}\left(\frac{A(H)}{B(H)}-\mathbb{E}_{p(H)}\frac{A(H)}{B(H)}\right)^2=0 \tag{29}$$

for any distribution $p(H)$. As a result, we have

$$\lim_{a\to\infty} \text{Var}_{p(H)}\left(\frac{A(H)}{B(H)}\right) = \lim_{a\to\infty} \int \left(\frac{A(H)}{B(H)} - \mathbb{E}_{p(H)}\frac{A(H)}{B(H)}\right)^2 p(H)dH$$

$$\leq \lim_{a\to\infty} \sqrt{\int \left(\frac{A(H)}{B(H)} - \mathbb{E}_{p(H)}\frac{A(H)}{B(H)}\right)^2 dH \int p(H)dH} = \sqrt{\lim_{a\to\infty} \int \left(\frac{A(H)}{B(H)} - \mathbb{E}_{p(H)}\frac{A(H)}{B(H)}\right)^2 dH} \ .$$

Because $\frac{A(H)}{B(H)}$ is bounded, it satisfies the conditions of Dominate Convergence Theorem, *i.e.*, the integration and limit operations are exchangeable, thus

$$\lim_{a\to\infty} \text{Var}_{p(H)}\left(\frac{A(H)}{B(H)}\right) \leq \int \lim_{a\to\infty}\left(\frac{A(H)}{B(H)} - \mathbb{E}_{p(H)}\frac{A(H)}{B(H)}\right)^2 dH = 0 \ .$$

On the other hand, we have $\lim_{a\to\infty} \text{Var}_{p(H)}\left(\frac{A(H)}{B(H)}\right) \geq 0$. As a result, we have

$$\lim_{a\to\infty} \text{Var}_{p(H)}\left(\frac{A(H)}{B(H)}\right) = 0 \ .$$

Substitute this into (19), one can obtain,

$$\lim_{a\to\infty} \mathbb{E}x_t x_{t+1} = \lim_{a\to\infty} \int \left[\frac{A(H)}{B(H)}\right]^2 p(H)dH$$

$$= \lim_{a\to\infty}\left(\int \frac{A(H)}{B(H)}p(H)dH\right)^2 + \lim_{a\to\infty} \text{Var}_{p(H)}\left(\frac{A(H)}{B(H)}\right)$$

$$= \lim_{a\to\infty}\left(\int \frac{A(H)}{B(H)}p(H)dH\right)^2 + 0 = \lim_{a\to\infty} \frac{1}{\Gamma(a)Z_1} \times \frac{[\int \frac{A(H)}{B(H)}\cdot B(H)e^{-H}dH]^2}{\int B(H)e^{-H}dH} \ ,$$

By changing the integration order, one can obtain, $\int A(H)e^{-H}dH = \Gamma(a)\int xe^{-U(x)}dx$. Similarly, $\int B(H)e^{-H}dH = \Gamma(a)\int e^{-U(x)}dx$. Notice that $Z_1 = \int e^{-U(x)}dx$,

$$\lim_{a\to\infty} \mathbb{E}x_t x_{t+1} = (\mathbb{E}x)^2,$$

Thereby, $\lim_{a\to\infty}\rho_x(1) = 0$ ∎

### E.3 Discussions for effective sample size

Effective sample size is associated with the variance of estimator based on MCMC sample [18], and can be used to measure the mixing performance of certain sampler. We hope to show that the ESS will become full sample size, indicating that the limiting behavior of Monte Carlo samples from analytic MG-SS becomes decorrelated, as $a$ approaches infinity. ESS is defined as ESS $= N/(1+2\times\sum_{h=1}^{\infty}\rho_x(h))$, where $N$ is the total number of samples, $\rho_x(h)$ is the $h$-step autocorrelation function. In this section, we first prove that $\rho_x(h)$ is non-negative. Then, assume the MG sampler is uniformly ergodic, *i.e.*, the total variance distance between the $h$-th transition kernel and $p(x)$ is bounded by $M(x)t^h$, where $M(x)$ is a bounded function and $0 < t < 1$ [32], under the condition that $\text{Var}_{\kappa_h(x_0|x)}x$ is bounded, where $\kappa_h(x_{t+h}|x_t)$ represents the $h$-*order transition kernel*, we can show that $\rho_x(h)$ is bounded by $Ct^{h/2}$, with $C$ a positive constant. If we further assume that $\rho_x(h)$ is monotonically decreasing, it can be shown that $\lim_{a\to\infty} \text{ESS} = N$. When ESS approaches full sample size, $N$, the resulting sampler delivers excellent mixing efficiency [5].

#### E.3.1 Proof of $\rho_x(h) \geq 0$

The $h$-time-lag autocorrelation function $\rho_x(h)$ can be formulated as

$$\rho_x(h) = \frac{\mathbb{E}_{p(x)}[\mathbb{E}_{\kappa_h(x_{t+h}|x)}x_{t+h}x] - (\mathbb{E}x)^2}{\text{Var}(x)} \ , \tag{30}$$

where, $\kappa_h(x_{t+h}|x_t)$ represents the *h-order transition kernel*, which can be calculated in a recursive manner as:

$$\kappa_1(x_{t+1}|x_t) = \int p(x_{t+1}|y_t)p(y_t|x_t)dy_t \,,\, \kappa_h(x_{t+h}|x_t) = \int \kappa_{h-1}(x_{t+1}|x_t)\kappa_1(x_{t+h}|x_{t+1})dx_{t+1} \,. \tag{31}$$

**Proposition 12** *The h-order transition kernel, $\rho_x(h)$, is non-negative.*

**Proof** From the reversibility shown above, we have,

$$\mathbb{E}x_{2h}x_0 = \mathbb{E}_{p(x_h)}[\mathbb{E}_{\kappa_h(x_{2h}|x_h)}x_{2h}\mathbb{E}_{\kappa_h^{-1}(x_0|x_h)}x_0] \geq [\mathbb{E}_{p(x_h)}\mathbb{E}_{\kappa_h(x'|x_h)}x']^2 = (\mathbb{E}x)^2$$

$$\mathbb{E}x_{2h+1}x_0 = \mathbb{E}_{p(x_h,x_{h+1})}[\mathbb{E}_{\kappa_h(x_{2h+1}|x_{h+1})}x_{2h+1}\mathbb{E}_{\kappa_h^{-1}(x_0|x_h)}x_0]$$

$$\geq [\mathbb{E}_{p(x_h,x_{h+1})}\mathbb{E}_{\kappa_h(x'|x_h)}x']^2 = (\mathbb{E}x)^2$$

Thus, by definition, $\rho_x(h) \geq 0$ ∎

### E.3.2 Discussions for effective sample size

**Proposition 13** *(Convergence of moments) Suppose a MCMC sampler is Harris ergodic with invariant distribution $p(x)$. Let $\kappa_h(\cdot, x)$ denote the h-th transition kernel. Define $\hat{x}_h(x_0) \triangleq \mathbb{E}_{\kappa_h(x_0,\cdot)}x_h$ as the expected value of h-time lag sample. If the variance of transition kernel $Var_{\kappa_h(\cdot,x)}(x)$ is bounded, when $h \to \infty$, we have,*

$$\hat{x}_h(x_0) \triangleq \mathbb{E}_{\kappa_h(x_0,\cdot)}x_h \to \mathbb{E}x,$$

**Proof** From Harris ergodicity, there exists $h'$ so that for $\forall \epsilon > 0$ and $h \geq h'$

$$\int_{\mathcal{X}} |\kappa_h(\cdot, x) - p(x)|dx < \epsilon$$

From Cauchy's inequality, considering the Harris ergodicity, one can obtain the convergence of the first moment as,

$$|\mathbb{E}_{\kappa_h(x_0,\cdot)}x_h - \mathbb{E}x| \leq \int_{\mathcal{X}} |x| \cdot |\kappa_h(x_0, x) - p(x)|dx$$

$$= \int_{\mathcal{X}} |x| \cdot |\kappa_h(x_0, x) - p(x)|^{\frac{1}{2}} |\kappa_h(x_0, x) - p(x)|^{\frac{1}{2}} dx$$

$$\leq \sqrt{\int_{\mathcal{X}} x^2 |\kappa_h(x_0, x) - p(x)|dx} \sqrt{\int_{\mathcal{X}} |\kappa_h(x_0, x) - p(x)|dx} \leq S \times M(x_0)^{\frac{1}{2}} t^{\frac{h}{2}}$$

Where $S \leq Var_{\kappa_h(x_0,x)}x + Var_{p(x)}x + 2[\mathbb{E}_{p(x)}x]^2$. Thereby,

$$\lim_{h\to\infty} |\mathbb{E}_{\kappa_h(x_0,\cdot)}x_h - \mathbb{E}x| = 0$$

∎

We propose an assumption as below,

**Assumption 1** *(Expected 1-lag sample) The expected 1-lag sample $\hat{x}_1(x)$ lies in between the interval defined by $x_0$ and $\mathbb{E}x$*

$$\frac{\hat{x}_1(x) - \mathbb{E}x}{x - \mathbb{E}x} \in [0, 1) \tag{32}$$

If such assumption holds for 1-time lag, the conclusion can be extended to $h$-time lag using below Lemma

**Lemma 14** *(Transitivity) Assume eqn. (1) holds when $h = 1$, it holds for any $h \in \{2, \cdots\}$.*

**Proof** We consider using induction, if eqn. (1) holds for $h = 1$. Without losing generality, we assume $x_0 \geq \mathbb{E}x$, thus we have $0 \leq \hat{x}_{h-1}(x_t) - \mathbb{E}x \leq x_0 - \mathbb{E}x$, holds for any $x_0$,

$$
\begin{aligned}
\hat{x}_h(x_0) - \mathbb{E}x &= \mathbb{E}_{k_1(x_1|x_0)}\hat{x}_h(x_1) - \mathbb{E}x = \mathbb{E}_{k_1(x_1|x_0)}[\hat{x}_h(x_1) - \mathbb{E}x] \\
&\leq \mathbb{E}_{k_1(x_1|x_0)}[x_1 - \mathbb{E}x] = \hat{x}_1(x_0) - \mathbb{E}x
\end{aligned}
$$

∎

Note that $\mathbb{E}x_h x_0 = \mathbb{E}_{p(x_0)}\hat{x}_h(x_0)x_0$ and $\mathbb{E}_{p(x_0)}\hat{x}_h(x_0) = \mathbb{E}x$, one can validate that

$$
\rho_x(h) = \frac{\mathbb{E}_{p(x_0)}[\hat{x}_h(x_0) - \mathbb{E}x](x_0 - \mathbb{E}x)}{\text{Var}(x)} = \frac{1}{\text{Var}(x)}\int[\hat{x}_h(x_0) - \mathbb{E}x]C(x_0)dx_0
$$

Where $C(x_0) \leq (x_0 - \mathbb{E}x)p(x_0)$. assumption 1 guarantees that $x_0 - \mathbb{E}x$ and $\hat{x}_h(x_0) - \mathbb{E}x$, have same sign. Without loss of generality, we assume $x_0 \geq \mathbb{E}x$, thereby,

$$
\rho_x(h) \leq \frac{S \times t^{\frac{h}{2}}}{\text{Var}(x)}\int M(x_0)^{\frac{1}{2}}C(x_0)dx_0
$$

This indicates that the $\rho_x(h)$ is bounded by an exponentially fast decreasing function, at a speed of $\mathcal{O}\left(t^{\frac{h}{2}}\right)$, where $t \in (0, 1)$ is the decay rate of total variance distance between $\kappa_h(\cdot, x)$ and $p(x)$. As a result

$$
\text{ESS} = \frac{N}{1 + 2 \times \sum_{h=1}^{\infty} \rho_x(h)} \geq \frac{N\text{Var}(x) \times (1-t)^{1/2}}{\text{Var}(x)(1-t)^{1/2} + 2St^{1/2}\int M(x_0)^{\frac{1}{2}}C(x_0)dx_0}
$$

The monotonicity of $\rho_x(1)$ could possible be shown by using below Lemmas and assumption 1.

**Lemma 15** *(Relative distance) Assume $p(x)$ is a well-defined probability density function with expectation $\mathbb{E}x$, $f_h(x)$ are a family of function of $x$, parameterized by $h$. if,*

$$
\frac{f_h(x) - \mathbb{E}x}{x - \mathbb{E}x} \in [0, 1)\,, \ \mathbb{E}f_h(x) = \mathbb{E}x
$$

*We have,*

$$
\mathbb{E}f_h(x)f_{h'}(x) < \mathbb{E}xf_{h'}(x) \tag{33}
$$

**Proof**

$$
\begin{aligned}
\mathbb{E}f_{h'}(x)f_h(x) - \mathbb{E}xf_h(x) &= \mathbb{E}[f_{h'}(x) - \mathbb{E}x][f_h(x)] \\
&= \mathbb{E}[f_{h'}(x) - x][f_h(x) - \mathbb{E}x] + \mathbb{E}x[\mathbb{E}f_{h'}(x) - \mathbb{E}x] \\
&= \mathbb{E}[f_{h'}(x) - x][f_h(x) - \mathbb{E}x] < 0
\end{aligned}
$$

The last inequality holds because the $[f_{h'}(x) - x][f_h(x) - \mathbb{E}x] < 0$ for any $x$. ∎

Given above results, note from stationary assumption that $\mathbb{E}_x\hat{x}_h(x) = \mathbb{E}_x\mathbb{E}_{\kappa_h(x'|x)}x' = \mathbb{E}x' = \mathbb{E}x$. From Lemma 15, letting $f_{h'}(x_0) = \hat{x}_1(x_0)$ and $f_h(x_0) = \hat{x}_h(x_0)$, one can obtain $\mathbb{E}x_h x_0 \geq \mathbb{E}x_{h-1}x_0$ for $t > 1$. Thus, $\rho_x(h) \geq \rho(h-1)$

**Proposition 16** *(Monotonicity) Monotonicity for autocorrelation function can be established if assumption 1 holds. $\rho_x(h) \geq \rho(h-1)$*

As shown in previous sections, when $a \to \infty$, the $\rho_x(1) \to 0$. Suppose the $\rho_x(h)$ is monotonically decreasing, Together with above result that $\rho_x(h)$ decrease in exponential speed, one can conclude that ESS would converge to the full sample size $N$.

**Theorem 17** *(Limiting ESS) If 1) assumption 1 holds, 2) the variance of transition kernel $\text{Var}_{\kappa_h(\cdot,x)}(x)$ is bounded, 3) uniform ergodicity can be established. When $a \to \infty$, we have, $ESS \to N$*

**Proof** From Equation (33), for any given $a$, there exists a $H$, such that, $Mt^{-H/2} < \rho_x(1)$ for $\forall h > H$. ($M$ denote the constant of the bound function). Thus,

$$\lim_{a \to \infty} \sum_{h=1}^{\infty} \rho_x(h) = \lim_{a \to \infty} \sum_{h=1}^{H} \rho_x(h) + \lim_{a \to \infty} \sum_{h=H+1}^{\infty} \rho_x(h) \leq \sum_{h=1}^{H} 0 + \lim_{a \to \infty} \rho_x(1) \frac{Mt^{-H/2}}{1 - t^{-1/2}} = 0$$

Note that $\sum_{h=1}^{\infty} \rho_x(h) > 0$. This indicates, $\lim_{a \to \infty} \sum_{h=1}^{\infty} \rho_x(h) = 0$. Thus, ESS $\to N$. ∎

### E.4 MG-HMC mixing performance

The *analytical MG-HMC* (without integration error, with adequate evolving time) is expected to have the same theoretical property as the analytical MG-SS since they are derived from the same problem setup. However, the mixing performance of the two methods could differ significantly, especially when sampling from a multimodal distribution.

Suppose we are sampling from a bimodal distribution. There must exist a critical value $y_T$, such that when the slicing variable $y$ exceeds $y_T$, the slice interval $\mathbb{X}$ will have two disjoint components. The corresponding Hamiltonian, $H$, will also have a critical value $H_T$, below which there would be two closed Hamiltonian contours associated with the same energy. The nature of Hamiltonian dynamics only allows moving along a single contour, whereas the analytic MG-SS is able to sample from distributions with disjoint domain, *i.e.*, $\mathbb{X}$ having several disjoint components. As a consequence, the analytical MG-HMC is expected to be less efficient than its analytic MG-SS counterpart. In order to move across different modes, the sampler has to have a large Hamiltonian, $H \geq H_T$.

To characterize the performance gap between the analytic MG-SS and MG-HMC, we note that the marginal distribution of $H$ can be obtained as $p(H; a) = [H - U(x)]^{a-1} e^{-H} / [\Gamma(a)Z_1]$. Therefore,

$$\begin{aligned} P(H \leq H_T) &= 1 - \int_{H \geq H_T} \frac{\int_{U(x) \leq H} [H - U(x)]^{a-1} dx}{\Gamma(a)Z_1} \times e^{-H} dH \\ &= 1 - \frac{1}{Z_1} \int \int_{H \geq \max(U(x), H_T)} [H - U(x)]^{a-1} \times e^{-H} / \Gamma(a) dH dx \\ &= \frac{1}{Z_1} \int_{U(x) \leq H_T} \frac{\gamma(a, H_T - U(x))}{\Gamma(a)} \times e^{-U(x)} dx \;, \end{aligned}$$

where $\gamma(\cdot, \cdot)$ denotes the lower incomplete Gamma function. Note that $F(a, x) = \frac{\gamma(a,x)}{\Gamma(a)}$ is the cumulative distribution function of Gamma$(a, 1)$, thus is monotonically decreasing with $a$, and as $a \to \infty$, $F(a, x) \to 0$. This implies that when $a$ is large enough, the chances of reaching an energy level that restricts the traversing across modes can be arbitrarily small. Note that as in Section D, the mass parameter $m$ have no impact on the analysis. As a result, in theory the analytical MG-HMC with large value of $a$ is particularly advantageous for sampling multimodal distributions.

Figure 7: Left (sample space): critical value (red) of slicing variable $y$, above which the slice interval will be disjoint. Right (phase space): critical value (red) of the Hamiltonian $H$, above which the contour will have two disjoint components.

## F  Theoretical autocorrelations and ESS for 1D cases

For derivation conveniency we first introduce several additional denotations. Note that $H = -\log y$, denoting $g(x, H) = H - U(x), s.t. U(x) \leq H$ to be the kinetic energy function *w.r.t.* $x$

conditioning on Hamiltonian $H$. We further denoting $A(H) = \int_{U(x) \leq H} x \cdot g(x, H)^{a-1} dx$, $B(H) = \int_{U(x) \leq H} g(x, H)^{a-1} dx$. $\mathbb{E} x_t x_{t+1}$ can be rewritten as,

$$\mathbb{E} x_t x_{t+1} = \int \frac{e^{-H}}{\Gamma(a) Z_1} \cdot \frac{A(H)^2}{B(H)} dH$$

## F.1 Theoretical autocorrelation for sampling exponential distribution

For $U(x) = x/\theta, x > 0$, using the definition of $A(H)$ and $B(H)$ from above, one can derive from algebra that,

$$B(H) = \frac{\theta H^a}{a} \;,\; A(H) = \frac{H^{a+1} \theta^2}{a + a^2} \;,\; \rho_x(1) = \frac{\mathbb{E} x_t x_{t+1} - \theta^2}{\theta^2} = \frac{1}{a+1}$$

Follow similar derivation, one could validate that,

$$\rho_x(h) = \frac{1}{(a+1)^h} \;,\; \text{ESS} = \frac{Na}{a+2} \tag{34}$$

## F.2 Theoretical autocorrelation for sampling positive-truncated Gaussian

In positive-truncated Gaussian case, $U(x) = x^2, x > 0$, we have,

$$B(H) = \frac{H^{a-\frac{1}{2}} \sqrt{\pi} \Gamma(a)}{2 \Gamma \left(a + \frac{1}{2}\right)} \;,\; A(H) = \frac{H^a}{2a}$$

$$\rho_x(1) = \frac{\mathbb{E} x_t x_{t+1} - 1/\pi}{1/2 - 1/\pi} = \frac{1}{(\pi/2 - 1)} \left[ \frac{\Gamma \left(a + \frac{1}{2}\right) \Gamma \left(a + \frac{3}{2}\right)}{\Gamma(a+1)^2} - 1 \right]$$

Thereby

$$\rho_x(1) = \frac{1}{(\pi/2 - 1)} \left[ \frac{\Gamma \left(a + \frac{1}{2}\right) \Gamma \left(a + \frac{3}{2}\right)}{\Gamma(a+1)^2} - 1 \right] \;,\; \rho_x(h) = \rho_x(1)^h \;,\; \text{ESS} = \frac{N}{1 + 2\rho_x(1)/(1 - \rho_x(1))}$$

## F.3 Theoretical autocorrelation for $U(x) = x^\omega$

The exponential family class of model introduced in [19] have the potential energy with form $U(x) = x^\omega, x \geq 0, \omega > 0$. For these model we have,

$$A(H) = \frac{H^{a-1+2/\omega} \Gamma(2/\omega) \Gamma(a)}{\omega \Gamma(a + 2/\omega)} \;,\; B(H) = \frac{H^{a-1+1/\omega} \Gamma(1/\omega) \Gamma(a)}{\omega \Gamma(a + 1/\omega)}$$

$$\rho_x(1) = \frac{\left[ \frac{\Gamma(a+3/\omega) \Gamma(a+1/\omega)}{\Gamma(a+2/\omega) \Gamma(a+2/\omega)} - 1 \right] \times \frac{\Gamma(2/\omega)^2}{\Gamma(1/\omega)^2}}{\frac{\Gamma(3/\omega)}{\Gamma(1/\omega)} - \frac{\Gamma(2/\omega)^2}{\Gamma(1/\omega)^2}}$$

A rough estimation for above $\rho_x(1)$ using Stirling's formula shows that $\rho_x(1) = \mathcal{O}(1/(a+1))$. Note that this results holds for $U(x) = x^\omega, x \geq 0, \omega > 0$, where $\theta$ is a scale parameter. Interestingly, [16] showed that if the integration time (leap-frog steps) is fixed, the geometric ergodicity holds only when $\omega \in [1, 2]$. However, with a random integration time the geometric ergodicity can be established for any $\omega > 0$. For this reason, we use a random integration time in our experiments.

## F.4 Theoretical autocorrelation for Gamma

We conducted numerical theoretical analysis on $\text{Gamma}(r, 1)$, where $r = 2, 3$. For each $a$, one can apply numerical methods for calculating the integrals $A(H)$ and $B(H)$. The $\rho_x(1)$ can then be calculated from Equation (18). The continuous function is plotted by interpolating from functional evaluation at $\{0.5, 1, 1.5, 2, 2.5, 3, 3.5, 4\}$ using quadrature.

# G    Remedy strategies for numerical issue and convergence issue

## G.1    Remedy strategies for numerical issue

To ameliorate numerical difficulties associated with large $a$ (Figure 8 (left)), we propose two remedies, i.e. **Reflection** and **Softened kinetics**.

Figure 8: Left: issues of large $a$. Large $a$ leads to a "stiffer" Hamiltonian trajectory. Right: Soft kinetic vs stiff kinetic

**Reflection**: Reflection performs well in low-dimensional phase space, but may suffer from sticky behavior in high-dimensional cases. This is because the probability of a sign change occurrence is high; recall there are at least $2^D$ turnovers. Besides, the transformation $(x, p) \mapsto (x, -p)$ depends on current $(x, p)$ and the determinant of the corresponding Jacobian is not one. This may lead to discrepancy in the sampled distribution, though the empirical discrepancy is not very large.

**Softened kinetics**: We define softened kinetics as

$$K(p) = -g(p) + 2/c \log(1 + e^{cg(p)}), g(p) = \text{sign}(p)|p|^{1/a}/m. \tag{35}$$

Where $c$ is a softening parameter. The comparison between standard kinetics ("stiff" kinetics) and softened kinetics are shown in Figure 8 (right). This kinetic share same tail behavior with stiff kinetic $K(p) = p^{1/a}/m$, and is differentiable, rendering much less numerical error. It asymptotically approaches standard kinetics when $c \to \infty$. One can use dimensional-wise importance sampling to correct the monomial gamma distribution to get momentum from the distribution defined by softened kinetics. However, when dealing with higher dimensional problems, the rejection rate of importance sampling step is high ($\mathcal{O}(D)$), which brings additional computational concerns.

In our tested scenarios, softened kinetics usually performs better than reflection in low dimensional problems, but fails to outperform reflection in high dimensional problems. For specific application, we advocate to try both to get the maximum of performance. In addition, note that for softened kinetics 1) the kinetics is modified, thus the theory can not be directly applied. 2) Softened kinetics requires heavier computation than reflection. 3) The parameter $c$ requires tuning.

Extensions that account for geometric information [5] or using more accurate numerical integrator [6] may also help alleviating the numerical problems.

## G.2    Remedy strategies for convergence issue

If the sampler is initialized in the tail region of a light-tailed target distribution, MG-HMC with $a > 1$ may converge arbitrarily slow to the true target distribution, i.e., the burn-in period could take arbitrarily long time. In Figure 9 we show this issue. To avoid being arbitrarily slow convergence with random initialization scheme, we suggest two strategies. First, we suggest using a step-size decay scheme, e.g., $\epsilon = \max(\epsilon_1 \rho^t, \epsilon_0)$. In our experiments we use $(\epsilon_1, \rho) = (10^6, 0.9)$, where $\epsilon_0$ is problem-specific. This allows the sampler to move larger to avoid slow convergence within burnin-steps, then gradually decreases to a normal step-size to perform stationary sampling. This approach empirically alleviates the slow convergence problem in our tested scenarios. Second, we suggest to initialize the sampler from a local maximum of posterior estimated from optimization methods, such as gradient descent method. This strategy ensure the sample is not initialized in light-tailed region. Third, with fixed computational budget, we encourage using reducing the leapfrog step in each iteration and increase the total number of iterations, which is essentially increasing the

Figure 9: The Hamiltonian trajectory when $a = 0.5$ (upper) and $a = 2$ (lower). When $a > 1$, the numerical difficulty increases, and samples of $x$ may move slowly in the light-tailed region (far from $x = 0$).

*resampling rate* of momentum variables. However, we note that the MG-HMC sampler with large $a$ may still be less sensitive when sampling from the light-tail, where a more sophisticated methods adaptively selecting $a$ during the sampling procedure are presumably useful. It can be proved that this adaptively selecting $a$ will still leave invariante target distribution. Nevertheless, we left this for further investigation.

## H Analytical MG-SS

### H.1 Analytical MG-SS for exponential distribution

For sampling an exponential distribution $\text{Exp}(\theta)$, *i.e.* $U(x) = \theta x, x > 0$, analytic MG-SS is available for all $a$. The procedure is given by

---
**Algorithm 3:** Analytical MG-SS for exponential distribution

    **Input**: Total sample size $S$
    **Output**: Sample results
    **Initialization:** Choose initial sample point $x_0$
    **for** $t = 1 \text{to} S$ do **do**
        Sample $K_t \sim \text{Gamma}(a, 1)$, find $H_t = x_t/\theta + K_t$
        Sample $\tau \sim \text{Uniform}(0, 1)$
        Sample $x_{t+1} = (1 - \tau^{1/a})\theta H_t$
    **end for**

---

### H.2 Analytical MG-SS for positive-truncated Gaussian distribution

Here we provide the algorithms for MG-SS in sampling positive-truncated Gaussian, *i.e.* $U(x) = x^2, x > 0$, for $a = 0.5, 1, 2$

---
**Algorithm 4:** Analytical MG-SS for half-Gaussian, $a = 0.5$

    **Input**: Total sample size $S$
    **Output**: Sample results
    **Initialization:** Choose initial sample point $x_0$
    **for** $t = 1 \text{to} S$ do **do**
        Sample $p_t \sim N\left(0, 1/\sqrt{2}\right)$, find $H_t = x_t^2 + p_t^2$
        Sample $\tau \sim \text{Uniform}(0, \pi)$
        Sample $x_{t+1} = \text{abs}\left(\sqrt{H_t}\cos(\tau)\right)$
    **end for**

---

For $a = 1$, this is standard slice sampler.

---
**Algorithm 5:** Analytical MG-SS for half-Gaussian, $a = 1$
---
    **Input**: Total sample size $S$
    **Output**: Sample results
    **Initialization:** Choose initial sample point $x_0$
    **for** $t = 1$to$S$ do **do**
        Sample $p_t \sim \text{Gamma}(1, 1)$, find $H_t = x_t^2 + p_t$
        Sample $\tau \sim \text{Uniform}(0, 2\sqrt{H_t})$
        Sample $x_{t+1} = \left(\tau - \sqrt{H_t}\right)$
    **end for**
---

For $a = 2$, we solve a cubic function to estimate $\mathbb{X}$, the resulting procedure is:

---
**Algorithm 6:** Analytical MG-SS for half-Gaussian, $a = 2$
---
    **Input**: Total sample size $S$
    **Output**: Sample results
    **Initialization:** Choose initial sample point $x_0$
    **for** $t = 1$to$S$ do **do**
        Sample $K_t \sim \text{Gamma}(2, 1)$, find $H_t = x_t^2 + K_t$
        Sample $\tau \sim \text{Uniform}(0, 1)$
        Compute $C = \left(2\sqrt{\tau(\tau-1)} + 1 - 2\tau\right) H_t^{1/2}$
        Sample $x_{t+1} = \text{abs}\left(-\frac{1+\sqrt{3}i}{2}C - \frac{1-\sqrt{3}i}{2C}\beta\right)$
    **end for**
---

Table 3: 1D exponential distribution. "AR" denotes acceptance rate. SS denote analytical slice sampler.

|  | Th. $\rho_x(1)$ | Th. ESS | SS $\rho_x(1)$ | SS ESS | HMC $\rho_x(1)$ | HMC ESS | HMC AR | HMC time(s) |
|---|---|---|---|---|---|---|---|---|
| $a = 0.5$ | 0.67 | 6,000 | 0.6620 | 6,204 | 0.6711 | 6,069 | 0.99 | 30 |
| $a = 1$ | 0.50 | 10,000 | 0.4868 | 10,227 | 0.5218 | 9,773 | 0.99 | 32 |
| $a = 2$ | 0.33 | 15,000 | 0.3265 | 15,547 | 0.3777 | 14,028 | 0.98 | 31 |
| $a = 3$ | 0.25 | 18,000 | 0.2494 | 17,507 | 0.2741 | 17,488 | 0.95 | 31 |
| $a = 4$ | 0.20 | 20,000 | 0.2108 | 19,229 | 0.2555 | 17,775 | 0.92 | 30 |

Table 4: 1D positive-truncated Gaussian. "AR" denotes acceptance rate. SS denote analytical slice sampler.

|  | Th. $\rho_x(1)$ | Th. ESS | SS $\rho_x(1)$ | SS ESS | HMC $\rho_x(1)$ | HMC ESS | HMC AR | HMC time(s) |
|---|---|---|---|---|---|---|---|---|
| $a = 0.5$ | 0.4787 | 10,576 | 0.4736 | 10,705 | 0.4802 | 10,510 | 0.99 | 42 |
| $a = 1$ | 0.3120 | 15,731 | 0.3040 | 15,457 | 0.3061 | 15,595 | 0.99 | 41 |
| $a = 2$ | 0.1830 | 20,718 | 0.1770 | 21,468 | 0.1937 | 20,498 | 0.99 | 43 |
| $a = 3$ | 0.1293 | 23,132 | - | - | 0.1665 | 21,303 | 0.96 | 65 |
| $a = 4$ | 0.0999 | 24,552 | - | - | 0.1508 | 22,115 | 0.94 | 120 |

Table 5: MG-HMC results of Gamma distribution

| $r = 2$ | Th. $\rho_x(1)$ | $\rho_x(1)$ | ESS | $r = 3$ | Th. $\rho_x(1)$ | $\rho_x(1)$ | ESS |
|---|---|---|---|---|---|---|---|
| $a = 0.5$ | 0.4600 | 0.3523 | 10457 | $a = 0.5$ | 0.3729 | 0.2182 | 15507 |
| $a = 1$ | 0.3023 | 0.3008 | 15248 | $a = 1$ | 0.2030 | 0.1979 | 18416 |
| $a = 2$ | 0.1891 | 0.1838 | 20979 | $a = 2$ | 0.1290 | 0.1223 | 23486 |
| $a = 3$ | 0.1372 | 0.1684 | 21703 | $a = 3$ | 0.0931 | 0.1572 | 22106 |
| $a = 4$ | 0.1077 | 0.2430 | 19062 | $a = 4$ | 0.0728 | 0.2116 | 19541 |

# I Complimentary experimental results

## I.1 Simulation results for 1D toy cases

The 1D simulation results are summarized in Table 3(Exp) ,Table 4(half-Gaussian) and Table 5(Gamma).

The comparison of $\rho_x(1)$ is provided in Figure 11. The skewness for $\text{Exp}(\theta)$, $\text{Gamma}(2,\theta)$, $\mathcal{N}_+(0,\theta)$ and $\text{Gamma}(3,\theta)$ are $\{2, 1.41, 0.99, 1.15\}$, respectively. We observed that a large value of shape parameter of Gamma distribution would lead to a lower $\rho_x(1)$, as $a$ fixed. Informally, this seems suggest that the skewness of the target distribution would influence the behavior of MG samplers. A more skewed distribution tends to have higher autocorrelation $\rho_x(h)$, and lower ESS.

We compare the empirical $\rho_x(1)$ and ESS of the analytic MG-SS and MG-HMC with their theoretical values of each cases in Figure 10. In the Gamma distribution case, analytic derivations of the autocorrelations and ESS are difficult, thus we resort to a numerical approach to compute $\rho_x(1)$ and ESS.

In principle, as $a$ becomes larger, it would be desirable to choose a smaller $\epsilon$ to compensate for the numerical hardness. Empirically, the choice of $m$ and $\epsilon$ is dependent. A small value of $m$ would compensate the demand for choosing a small step-size to certain extent. However, optimal performances were achieved by tuning both of them. Presumably, $m$ will influence the relative scale of the contour along $x$-axis and $p$-axis, thus tuning $m$ will influence the general shape of the contour, which may be beneficial in some cases. For the exponential and positive-truncated Gaussian cases, as $a$ becomes larger, the autocorrelation decreases from 1 to a small value close to zero, meanwhile the ESS increases to approach the total sample size. However, the acceptance rates also decrease.

The results for analytic MG-SS match well with the theoretical results, however MG-HMC seems to suffer from practical difficulties when $a$ is large, evidenced by results gradually deviating from the theoretical values. This issue is more prominent in the Gamma case, where the autocorrelation first decreases then becomes larger.

Figure 10: Theoretical and empirical $\rho_x(1)$ and ESS of exponential distribution (a,b) and $\mathcal{N}_+$ (c,d). $\rho_x(1)$ for Gamma distribution with parameters $r = 2$ (e) and $r = 3$ (f)

Table 6: BLR setup (dimensionality of each dataset is indicated in parenthesis)

| Dataset(dim) | Australian(15) | German(25) | Heart(14) | Pima(8) | Ripley(7) | Cavaran (87) |
|---|---|---|---|---|---|---|
| $\epsilon$ | 0.1 | 0.05 | 0.14 | 0.1 | 0.14 | 0.03 |
| $m$ when $a = 0.5$ | 10 | 10 | 10 | 10 | 10 | 10 |
| $m$ when $a = 1$ | 2 | 2 | 2 | 2 | 2 | 2 |
| $m$ when $a = 2$ | 1 | 1 | 1 | 1 | 1 | 1 |

Figure 11: Comparison of $\rho_x(1)$ for 4 simulated univariate cases

Table 7: Experiment setups for 1D simulated study

| Exponential | $m$ | $\epsilon$ | AR | Gaussian(+) | $m$ | $\epsilon$ | AR |
|---|---|---|---|---|---|---|---|
| $a = 0.5$ | 1 | 0.05 | 0.991 | $a = 0.5$ | 1 | 0.05 | 0.996 |
| $a = 1$ | 1 | 0.05 | 0.982 | $a = 1$ | 1 | 0.1 | 0.981 |
| $a = 2$ | 0.15 | 0.05 | 0.980 | $a = 2$ | 0.15 | $5 \times 10^{-3}$ | 0.983 |
| $a = 3$ | 0.02 | $1 \times 10^{-3}$ | 0.94 | $a = 3$ | 0.02 | $5 \times 10^{-5}$ | 0.962 |
| $a = 4$ | $3 \times 10^{-3}$ | $5 \times 10^{-8}$ | 0.90 | $a = 4$ | $3 \times 10^{-3}$ | $2.5 \times 10^{-8}$ | 0.946 |
| Gamma, $r = 2$ | $m$ | $\epsilon$ | AR | Gamma, $r = 3$ | $m$ | $\epsilon$ | AR |
| $a = 0.5$ | 1 | 0.05 | 0.986 | $a = 0.5$ | 1 | 0.05 | 0.986 |
| $a = 1$ | 1 | 0.1 | 0.982 | $a = 1$ | 1 | 0.05 | 0.983 |
| $a = 2$ | 0.15 | $5 \times 10^{-3}$ | 0.965 | $a = 2$ | 0.15 | 0.01 | 0.972 |
| $a = 3$ | 0.02 | $2.5 \times 10^{-5}$ | 0.902 | $a = 3$ | 0.02 | $1 \times 10^{-5}$ | 0.885 |
| $a = 4$ | $3 \times 10^{-3}$ | $2.5 \times 10^{-8}$ | 0.783 | $a = 4$ | $3 \times 10^{-3}$ | $2.5 \times 10^{-8}$ | 0.766 |

Table 8: Effective sample size of MG-HMC for 1D and 2D bimodal distribution. "AR" denotes acceptance rates.

| 1D | ESS | $\rho_x(1)$ | AR | 2D | ESS | $\rho_x(1)$ | AR |
|---|---|---|---|---|---|---|---|
| $a = 0.5$ | 5175 | 0.60 | 0.98 | $a = 0.5$ | 4691 | 0.67 | 0.96 |
| $a = 1$ | 10157 | 0.43 | 0.97 | $a = 1$ | 16349 | 0.60 | 0.87 |
| $a = 2$ | 24298 | 0.11 | 0.92 | $a = 2$ | 18007 | 0.53 | 0.78 |

Table 9: Comparison between MG-HMC $a = 1$ with standard slice sampling in 1D unimodal toy cases

| MG-HMC($a = 1$) | Exponential | half-Gaussian | Gamma($r = 2$) | Gamma($r = 3$) |
|---|---|---|---|---|
| $\rho_x(1)$ | 0.5218 | 0.3061 | 0.3008 | 0.1979 |
| ESS | 9,773 | 15,595 | 15,248 | 18416 |
| Standard SS | Exponential | half-Gaussian | Gamma($r = 2$) | Gamma($r = 3$) |
| $\rho_x(1)$ | 0.5198 | 0.3039 | 0.3011 | 0.1954 |
| ESS | 9,622 | 16,051 | 15,092 | 18874 |

Table 10: Comparison between MG-HMC $a = 1$ with standard slice sampling in 1D and 2D bimodal toy cases

| MG-HMC($a = 1$) | 1D | 2D | Standard SS | 1D | 2D |
|---|---|---|---|---|---|
| $\rho_x(1)$ | 0.43 | 0.60 | $\rho_x(1)$ | 0.056 | 0.697 |
| ESS | 10157 | 16349 | ESS | 27469 | 9566 |

## I.2 Comparison between MG-HMC $a = 1$ with standard SS

**1D unimodal toy cases** To validate that when $a = 1$, the resulting sampler can be understood as standard slice sampling, we also compared with standard slice sampling using doubling and shrinking scheme [4]. The resulting ESS and $\rho(1)$ is almost identical to analytical MG-SS and MG-HMC with $a = 1$. The results are reported in Table 9.

Table 11: Comparison between MG-HMC $a = 1$ with standard slice sampling in BLR and ICA experiments

| Min ESS | A | G | H | P | R | C | ICA |
|---|---|---|---|---|---|---|---|
| MG-HMC($a = 1$) | 4308 | 4353 | 4591 | 4664 | 4226 | 36 | 3029 |
| Standard SS | 8.1 | 9.7 | 5.7 | 5.2 | 3.7 | 42.9 | 3.29 |

**Bimodal 1D and 2D cases** We also applied standard slice sampler with dimensional-wise doubling and shrinking scheme [4] for these bimodal tasks. In 1D case, the standard slice sampler yields ESS close to full sample size, while in 2D cases, the resulting ESS is 9533. This is reasonable because when sampling from these bimodal symmetric distribution, in theory, analytical MG-SS gives full ESS. However the slice sampler is usually less efficient when dealing with more than one dimension. We note that there are more sophisticated methods for performing slice sampling in multidimensional [11], however we leave the comparison for future investigation. The results are reported in Table 10.

**BLR and ICA** We reported the result for standard slice sampler [4] in Table 11. In general, standard slice sampler with adaptive search fails to achieve a comparable results with other compared methods when applying to multi-dimensional scenarios.

### I.3  100 dimensional multivariate Gaussian

We assessed the performance of MG-HMC for sampling 100 dimensional Gaussian distribution. The target Gaussian distribution has zero-mean and diagonal covariance matrix, where the diagonal elements are uniformly drawn from $[0, 10]$. We collected 5000 MC samples after applying 2500 burn-in rounds. We compared the efficiency and accuracy of MG-HMC with $a = \{0.5, 1, 2\}$. For each scheme, we use 5 different leapfrog step-sizes $\epsilon_t$, $t = \{1 \cdots 5\}$, where $\epsilon_{t+1} = 0.8\epsilon_t$, so as to make the acceptance rates ranges from 0.4 to 0.9. MG-HMC with $a = 1$ achieved highest median effective sample size, as well as lowest mean square error between empirically estimated parameters and truth (Figure 12). The maximum acceptance rates for $a = \{0.5, 1, 2\}$ is $\{0.98, 0.96, 0.63\}$, respectively. MG-HMC with $a = 2$ failed to outcompete other two tested schemes, probably due to the increasing numerical hardness.

Figure 12: Scatter plot of mean squared error of estimated covariance and median ESS for simulated 100D Gaussian distribution. Number labels denote the stepsize index.

## J  Experimental setup

**1D toy synthetic problems** we use a random integration time (leap-frog steps) uniformly drawn from $(20, 180)$, which has better convergence guarantee as suggested by [16]. Step sizes and $m$ are selected such that the acceptance rates fall within $[0.6, 0.9]$, as suggested by [33]. The parameters for MG-HMC in our simulation study is selected by grid search. Specifically, we tried stepsize $\epsilon \in (0.05, 0.025, 1 * 10^{-2}, \cdots, 10^{-8})$, and mass parameter $m \in 2, 1, 0.5, 0.25, 0.15, 0.05, 0.02, 0.003$. For univariate distributions the optimal setup is provided in the Table 7.

**Simulated bimodal experiments** Each leap-frog update has $(50 - l, 50 + l), l = 20$ steps, the step-size is set as $\epsilon = 0.05$. The mass parameter for 1D case is chosen to be $m = \{5, 1.2, 0.4\}$ for $a = \{0.5, 1, 2\}$, respectively. For 2D case, the mass matrix is obtained by a mass parameter $m$ times the identity matrix, where $m = \{1, 0.1, 0.35\}$ for $a = \{0.5, 1, 2\}$.

**Bayesian logistic regression** We follow the setup in [5] and [6] for BLR experiment. For data $\mathbf{X} \in \mathbb{R}^{d \times N}$, response variable $\mathbf{t} \in \{0, 1\}^N$ and target parameters $\boldsymbol{\beta} \in \mathbb{R}^d$, if we impose a Gaussian

Table 12: The avg AUROC for each method. Dimensionality of each dataset is indicated in parenthesis after the name of each dataset.

| Dataset ($D$) | Australian (15) | German (25) | Heart (14) | Pima (8) | Ripley (7) | Cavaran (87) |
|---|---|---|---|---|---|---|
| $a = 0.5$ | 0.92 | 0.78 | 0.92 | 0.90 | 0.89 | 0.82 |
| $a = 1$ | 0.93 | 0.79 | 0.93 | 0.88 | 0.86 | 0.84 |
| $a = 2$ | 0.92 | 0.79 | 0.96 | 0.94 | 0.87 | 0.69 |

prior $\mathcal{N}(\mathbf{0}, \alpha\mathbf{I})$ (where $\alpha > 0$) on $\boldsymbol{\beta}$, the log posterior is given by [5],

$$\mathcal{L}(\boldsymbol{\beta}) = \boldsymbol{\beta}^T\mathbf{X}\mathbf{t} - \sum_{n=1}^{N}log(1 + \exp(\boldsymbol{\beta}^T\mathbf{X}_{n,\cdot}^T)) - \frac{\boldsymbol{\beta}^T\boldsymbol{\beta}}{2\alpha}$$

We collect 5000 MC samples, with 1000 burn-in samples. For each dataset, we use a random integration time (leap-frog steps) uniformly drawn from $(20, 180)$, which has better convergence guarantee as suggested by [16]. Step sizes and $m$ are selected such that the acceptance rates fall within $[0.6, 0.9]$, as suggested by [33]. The stepsize and mass parameter varies from dataset to dataset (Table 6). To deal with numerical problems, in Table 6, for $a = 1$ we use reflection, for $a = 2$ we use softened kinetics. The softening parameter $c$ is set as $[0.3, 0.2, 0.2, 0.2, 0.3, 0.2]$ for the 6 datasets, respectively.

The average AUROC based on 10 folds cross-validation for each method is reported in Table 12

**Independent component analysis** For data $\mathbf{X} \in \mathbb{R}^{d \times N}$ and target parameters $\mathbf{W} \in \mathbb{R}^{d \times d}$, the joint likelihood is given by [34, 6],

$$p(\mathbf{X}, \mathbf{W}) = |\det(\mathbf{W})|^N \prod_{i=1}^{N}\prod_{j=1}^{d} p_j(w_j^T x_i) \prod_{k,l} \mathcal{N}(W_{kl}; 0, \sigma)$$

In our experiments, we set the variance of the Gaussian prior to 100. The $p_j(w_j^T x_i) = \{4cosh^2(1/2y_{ij})\}^{-1}$, where $\mathbf{y}_i = \mathbf{W}\mathbf{x}_i$ [6, 35].

We collect 5000 MC samples, with 1000 burn-in samples. The setups are provided in (Table 13). The number of leap-frog steps are uniformly drawn from $(20, 180)$. The computational time is almost identical, $(525, 517, 523)$ seconds for $a = (0.5, 1, 2)$, respectively.

Table 13: ICA setup

| ICA | $m$ | $\epsilon$ | AR |
|---|---|---|---|
| $a = 0.5$ | 2 | $1.5 \times 10^{-5}$ | 0.986 |
| $a = 1$ | 1.7 | $5 \times 10^{-5}$ | 0.973 |
| $a = 2$ | 0.5 | $1 \times 10^{-5}$ | 0.772 |