[Reviews · NeurIPS 2016]

Reviewer 1

Summary

The authors link HMC method with the slice sampling technique. This study allows them to design a novel HMC algorithm (they employ a non-Gaussian kinetic distribution).

Qualitative Assessment

Strength: I believe that finding connections between different computational methods is a very important task in the modern research scenario. These studies require a great effort and dedication of the authors, and they are really useful, in my opinion. Thus, I enjoy to read Section 3. In this sense, this work deserves to be published, in my opinion. Weakness: in my opinion, HMC is a nice idea but I do not understand the huge amount of extensions that can be found in literature. I believe that most of them are unnecessary complicated and not really competitive with respect to (gradient-free) advanced MCMC techniques (Multiple try schemes, adaptive MCMC etc.). Moreover, it is difficult to find honest discussions about some drawbacks of HMC (selection of \tau, behavior with highly multimodal target- 10 modes for instance etc.). Hence, I am not sure that the novel proposed technique is really a relevant improvement. This aspect should be clarified (explained in a more intuitive way).

Confidence in this Review

2-Confident (read it all; understood it all reasonably well)


Reviewer 2

Summary

This paper attempts to draw a tight connection between two well-documented Markov chain Monte Carlo (MCMC) sampling algorithms, namely Hamiltonian Monte Carlo and slice sampling. By reformulating the standard Hamiltonian dynamics via the Hamilton-Jacobi equation, the authors exhibit a HMC sampling scheme (HMC-SS) similar to slice sampling. Then going one step further allows the authors to derive a generalized HMC scheme by assigning the momentum variable a so-called monomial Gamma (MG) distribution (which is nothing but a particular instance of double-sided Gamma distribution) as prior distribution. This resulting MG-HMC algorithm generalizes the standard HMC and slice sampling.

Qualitative Assessment

*Formulating HMC as a slice sampler* Reformulating standard slice sampler from HMC-SS requires to assign the momentum variable a monomial Gamma distribution, which leads to non-standard instances of the corresponding kinetic functions. It would have been interesting to discuss these kinetic function and its interpretation with respect to the Hamiltonian dynamics framework. More precisely, in standard instance of HMC, the momentum variables are associated with a the so-called "kinetic" function characterized by its quadratic form, which underlies the velocity interpretation of these momentum variables. When the momentum prior is chosen as monomial Gamma distribution, can we still refer to K as a kinetic function? *Experiments* 1. When investigating the mixing properties of the proposed MG-SS and MG-HMC samplers, the authors state that "The acceptance rates decrease from around 0.98 to around 0.77 for each case, when a grows from 0.5 to 4, as shown in Figure 2(a)-(d)." However, unless the reviewer missed something, these acceptance rates are not depicted in these plots. 2. According to the reviewer's point of view, the experiments do not really emphasize the actual usefulness of the proposed sampling schemes. The distributions considered in the simulation studies are rather standard and the problem considered with real data could be probably tackled using more efficient strategies. Instead, the authors could have concentrated their efforts on sampling from more challenging distributions, in particular when they are constrained on challenging sets such as considered in [Betancourt2012]. [Betancourt2012] M.J. Betancourt, "Cruising The Simplex: Hamiltonian Monte Carlo and the Dirichlet Distribution", in Proc. Maxent, 2012. *Other minor comments* Please use capital letters when required, in particular in the bibliographic section (Hamiltonian, Riemann, Monte Carlo...).

Confidence in this Review

2-Confident (read it all; understood it all reasonably well)


Reviewer 3

Summary

This paper derives a framework in which a type of HMC and slice sampling form special cases (in one dimension), motivating a family of HMC samplers with generalized Gaussian momentum distributions, which are shown similarly to be associated to particular generalizations of slice samplers.

Qualitative Assessment

The connection between these two separate auxiliary variable Monte Carlo methods is interesting; some previous work has been done in this general area (e.g. Slice Sampling on Hamiltonian Trajectories, ICML 2016), although the specific connection presented in this paper is novel as far as I’m aware. Overall the quality of the presentation is fairly good. One point I found confusing was that the general set-up for MG-HMC and MG-SS (including explicit statements of the algorithms) is given in Section 3 entirely for the one-dimensional case. In Section 5 and beyond, the methods are used in multidimensional cases, without much explanation given as to how the algorithms/derivations as stated in the previous sections adapt for more than one dimension (e.g. when Hamiltonian trajectories are not necessarily closed) - perhaps it could be made explicit in the main paper exactly how the samplers generalize to more than one dimension? Some description of how this generalizes to more than one dimension is given in the appendix, although there is relatively little detail and/or justification of the points made here as far as I can see - i.e. what happens if the potential isn’t decomposable across dimensions, and what does “hyper-diamond” mean in the footnote of p10? It’s also not clear to me what is meant by “almost certainly” in line 457 - there are cases where the trajectories are not space-filling (e.g. Gaussian kinetic and potential energies). The theoretical results in the main paper are clearly presented, and demonstrate a nice property of the one-dimensional analytic MG slice sampler for a large class of potential functions. However, there appears to be some discrepancy between the results as stated in the main paper, and the assumptions used in the proofs in the appendix. For example, it seems to be the case that the chain is assumed to be in its stationary distribution in order to prove Theorem 1, but this is not mentioned in the main paper - it would be good to get clarification from the authors on this point, and whether there are any other conditions needed to obtain the results as stated in the main text. The analytic toy example of the exponential distribution is a nice addition that demonstrates some of the theoretical results explicitly in a particular case. Again, it is not clear to me from the paper whether these results would generalize to more than one dimension. It is mentioned in the paper that when the slice region is disconnected, there may be some discrepancy between the performance of analytic MG-HMC and analytic MG-SS, although it is not made clear whether the ACF results earlier in the section would be expected to generalize. It’s also not clear to me whether these results generalize to multidimensional versions of the samplers. My two main concerns about the paper are: i) whether the set-up of Sections 2 and 3 is particular to the one-dimensional case, and whether the multi-dimensional implementation of MG-HMC described in Section 5 bears much relation to the analytic MG-HMC samplers, and generalized slice samplers, described earlier in the paper; ii) what is the full list of conditions needed to ensure the validity of some of the results in the appendix, and whether the statements of results in Section 4 of the main paper reflects these conditions. The appendix appears to contain quite a few typos that should be corrected e.g. line 455 “If uniformly sample”, line 464 “in each of the case”, line 465 “in cases more than one dimension” etc. With regards to experiments, the simulation studies provide a nice illustration of the one-dimensional theoretical results discussed earlier, and indicate where numerical issues cause practical performance to deviate from the theoretical expectation. The Bayesian logistic regression experiment also demonstrates that improved mixing can be achieved when taking a value of the parameter "a" not equal to 0.5, which nicely demonstrates that the general MG-HMC framework introduced yields samplers that can out-perform HMC with traditional Gaussian momentum. Minor points: Line 166: MG-SS mentioned before being introduced - possible typo? Line 259: In the second equation, I think there is a missing sign(p)_d on the right-hand side.

Confidence in this Review

2-Confident (read it all; understood it all reasonably well)


Reviewer 4

Summary

The paper reveals a framework of generalized slice sampling that both the conventional slice sampling and HMC (by expressing Hamiltonian dynamics with HJE) can fit in, thus bridges HMC and slice sampling. Furthermore, the paper shows that slice samplers in this framework can be implemented by both slice sampling and Monomial Gamma Hamiltonian Monte Carlo, which motivates MG-HMC, an HMC-implemented (conventional) slice sampler, which is better than traditional HMC in lower sample autocorrelation (more efficient moves). Theoretical analysis on asymptotic properties of instances of the framework is provided and results of both synthetic and real-data experiments support the conclusions of the analysis.

Qualitative Assessment

- Technical quality: the authors have a very solid background on classical mechanics and probability theories. However, there are still some issues that seems unclear to me: 1) The vanilla HMC uses a fixed time interval for evolving the dynamics (i.e. $\tau_t$ is constant and does not depend on $t$) where the proposal $\{x_t(\tau_t), p_t(\tau_t)\}$ is generated by a deterministic mapping from $\{x_t(0), p_t(0)\}$ with a unit Jacobian. So MH test is not needed if exactly simulated. But for the modified HMC with uniformly randomly drawn time interval, which is the leading role of the paper, generates a proposal by drawing a sample from some distribution, whose marginal density of $x$ is given in Equation (6). So should an MH test be added even the dynamics is exactly simulated? Since Algorithm 1 is equivalent to generating proposals in such a way (exact simulation), does it need to be corrected by an MH test? 2) Is the procedure for sampling a slice (Equation (7)) with $a=1/2$ equivalent to momentum resampling from Gaussian? If yes, $y_t$ should be sampled uniformly randomly from the inteval $(0, f(x_t))$, but the density in Equation (7) seems not constant. If no, then how to understand Equation (7)? There are some other minor issues: 1) The way to write Hamiltonian as $H(x, p, \tau)$ in Line 53 is confusing. In my opinion, it should be written as $H(x, p, \tau) = K(p)+U(x)$ or $H(x, p) = K(p)+U(x)$ or $H(\tau) = K(p(\tau))+U(x(\tau))$. If written in $H(x, p, \tau)$ as the form of a general Hamiltonian, $H$ depends on time $\tau$ explicitly but not through $x$ or $p$, and the time should be regarded as a variable independent of $x$ and $p$. For the most common specific form of Hamiltonian, $H$ is independent of $\tau$ explicitly. In this case the Hamiltonian can be written as $H(x, p)$. If a specific trajectory ${x(\tau), p(\tau)} of the dynamics is given, the Hamiltonian on this trajectory is a function purely of time $\tau$, where we can write the third form. However, to specify a dynamics, Hamiltonian should be written as $H(x, p, \tau)$ or $H(x, p)$ if it does not depend on $\tau$ explicitly. 2) $S$ can be any differentiable function in Line 72. If we force the transformed Hamiltonian $H'$ to be constant 0, then $S$ is specified as Hamilton's principal function. - Novelty/originality: it is very impressive to reveal the connection between HMC and slice sampling, the two seemingly quite different samplers. Besides, by casting the two samplers into one general framework, unified analysis and comparison can be carried out for them, e.g. HMC is less efficient than conventional slice sampling in effective moves (sample autocorrelations). The connection also makes it possible to implement HMC via slice sampling and to implement slice sampling via simulating Hamiltonian dynamics, which motivates MG-HMC, an HMC-like sampler with comparable effective moves as conventional slice sampling. - Potential impact or usefulness: as stated, although of limited practical impact, the work helps to understand HMC in depth and the framework unifying HMC and slice sampling is promising to generate new samplers. But it may appear a little bit difficult for many people to digest. - Clarity and presentation: generally reads well. But there are some possible flaws: 1) In Line 166, the abbreviation "MG-SS" appears for the first time without a full name. 2) Algorithm 2 for MG-SS is not introduced. It is unclear how the algorithm is raised and what its relationship with Algorithm 1 is. - Possible typos: 1) In the last line of Algorithm 1, the exponent $a-1$ should be $1-a$. 2) In Line 216, $a\to\infty$ should be $t\to\infty$.

Confidence in this Review

2-Confident (read it all; understood it all reasonably well)


Reviewer 5

Summary

The authors make the somewhat unexpected (to me at least..) connection between HMC and slice sampling, by introducing a generalised kinetic function, a particular choice of parameters for which results in each of the methods. The resulting MG-HMC method then offers a new family of HMC algorithms, which the authors investigate theoretically and in practice.

Qualitative Assessment

The authors make the very interesting and unexpected connection between standard HMC and slice-sampling, showing them both to be member of a more general family of HMC methods constructed using a generalised kinetic function. This seems to be very novel and of great theoretical interest, although it is not clear to me that it results in any great practical benefit to the average practitioner. Some guidance on whether this is indeed the case, and where the use of MG-HMC is likely to improve performance, would be useful - the examples given could be more efficiently sampled from using alternative methods. Overall the paper was clearly presented and reasonably straightforward to follow.

Confidence in this Review

2-Confident (read it all; understood it all reasonably well)


Reviewer 6

Summary

The main ideas behind this work is to combine two major classes of auxiliary variable MCMC methods -- HMCs and slice sampling. The authors propose a way to reformulate slice sampling in an HMC like setting through generating samples for the momentum distribution. Specifically, this is done using the Monommial Gamma distribution which is a more general version of the exponential family (where Gaussians are specific instantiations of the MG when a=0.5).

Qualitative Assessment

I think this paper has some very interesting ideas. It provides a very rigorous overview and delves into the intricacies of HMC authoritatively. I found some of the proofs difficult to comprehend to authoritatively accept or reject but I believe the intuition behind them. A 28-page appendix is a little challenging for a reviewer to go over for a NIPS submission, but skimming through sections it seems like there is a lot of comparisons already exist. A criticism I have of this work is comparison to other existing methods and also standard HMC. I came away reading the paper and not knowing what obvious performance improvements/gains this method actually has. For example, HMCs are notoriously sensitive to the hyperparameters (step length, number of steps, momentum corruption) So it is entirely possible that two HMC methods perform very differently for the same hyper-parameters. A more accurate comparison would be to find optimal hyperparameters for each method and then test the performance. One way to do this would be to use a tool like Spearmint (HIPS, from Ryan Adams' group) to test convergence of a hyperparamter and then show results.

Confidence in this Review

2-Confident (read it all; understood it all reasonably well)